# Characteristics of Temperature Field of Rammed Earth Wall in Arid Environment

Qiangqiang Pei [1,2,3,4,*], Bo Zhang [1,2,3,4], Dongjuan Shang [2,3,4], Qinglin Guo [1,2,3], Jinjing Huang [1,2,3] and Jing Zhu [1,2,3]

[1] Dunhuang Academy, Mogao Grottoes, Dunhuang 736200, China; zb20418@dha.ac.cn (B.Z.); gql20183@dha.ac.cn (Q.G.); hjj20384@dha.ac.cn (J.H.); zj70105@dha.ac.cn (J.Z.)
[2] National Research Center for Conservation of Ancient Wall Paintings and Earthen Sites, Dunhuang 736200, China; 18368915808@163.com
[3] Key Laboratory for Conservation of Ancient Wall Paintings and Earthen Sites, Dunhuang 736200, China
[4] Gansu Mogao Grottoes Cultural Heritage Protection Design Consulting Co., Ltd., Dunhuang 736200, China
* Correspondence: peiqiangq@163.com

**Abstract:** The rammed earth walls were greatly affected by the change of temperature fields in arid environments, particularly the swell-shrink stress caused by gradient variations of the temperature fields was one of the main factors leading to surface weathering of rammed earth sites. While heat conduction, convection, and radiation were the main factors resulting in temperature variations of rammed earth walls. In this study, the thermocouple sensors were embedded in a rammed-earth test wall, to continuously monitor the temperature gradient field of the rammed earth wall from the near-surface area to the interior. The results revealed that the wall was greatly influenced by seasonal temperature fields together with significant changes in daily temperature differences. The top and the surface of the wall were affected by thermal radiation and convection, while the interior and the foundation were affected by heat conduction. The annual temperature difference reached 62.99 °C, and the maximum daily temperature difference on the surface of the wall was 24.3 °C, which mainly appeared during the spring and autumn seasons. The near-surface thickness of the temperature-sensitive area of the wall was less than 32 cm, within which the temperature varied dramatically at depths of 0–18 cm. The temperature variations at depths of 18–32 cm were somewhat significant with no obvious changes at depths exceeding 32 cm. These trends indicate that the rammed earth wall has an outstanding function of thermal preservation and insulation. This study is expected to be of guidance and reference for multi-field coupling environmental condition setting for surface weathering of rammed earth site bodies, stress and strain caused by temperature field, surface weathering mechanism and strengthening technology as well as the related researches of modern rammed earth building designs.

**Keywords:** rammed earth sites; temperature field; gradient law; arid environment

## 1. Introduction

In recent years, as human awareness of the environment has been improved and problems related to housing energy efficiency and environmental aspects have been concerned, rammed earth structures have attracted particular attention owing to their special features such as poor heat conduction, good thermal insulation, and low energy consumption [1,2]. The intense thermal energy transference in a certain range of depths from the surface of the rammed wall ensures relatively constant temperature and humidity for inside parts of buildings [3]. However, conservators of rammed earth sites have focused mostly on the damage from dramatic temperature fluctuations on the surfaces of such sites. Matthew Hall et al. believe that thermal conductivity materials increased linearly with their saturation ratio [4]. Zheng Long et al. adopt the method of implanting silicon semiconductor sensors in the wall to monitor the wall temperature and find that the temperature gradient

varied clearly from the wall surface to the center [5]. Globally, cultural and heritage sites are carefully preserved on all continents, and more than one-third of these sites are earthen sites [6,7]. A large number of culturally rich earthen sites are preserved in northwest China [8]. Under the influence of long-term natural and human factors, these sites have suffered from different types of deterioration [9]. The weather in northwest China is extremely variable, with the annual maximum temperature difference reaching 80 °C; the daily maximum temperature difference approaches 30 °C in some areas. Earthen sites in some locations are covered in snow for long periods during winter and directly exposed to sunlight in summer. Therefore, the temperature fields between the superficial and internal layers of these sites vary greatly, thereby causing long-term repeated expansions and contractions that can gradually affect the structures of the superficial layers [10]. This is the main driving force for the intensification of surface weathering, flaking, stripping, and foundation undercutting damages in earthen sites (Figure 1). Jenny Richards et al. established a preliminary model for sand degradation of earthen sites in the arid area [11]. Du Yumin et al. took the Great Wall of Qinghai as the object to evaluate the damage degree of the sites by means of a neural network and fuzzy analytic hierarchy process and ranked the main influencing factors [12,13]. Although the influence of the natural environment on the destruction of earthen sites is a multi-field coupling effect, the main causes are considered to be water and temperature variations. Fodde, Enrico et al. consider that groundwater is the key driver for salt-eroded sites [14]. Sun Manli and Susana Serrano, etc. believe the rammed earth wall was easy to weather and fall off under the action of water and heat [15,16]. The changes in the temperature fields at the sites are greatly affected by the seasonal climate, which is the necessary driving force for the movement of water and salt as well as the expansion and contraction of the earthen sites; these variations also render the rammed earthen sites more prone to damages [17–19]. Recently, Shen et al. [20] studied the model of random solar radiation energy, especially the processes of storing and releasing energy, which can help reduce fluctuations in indoor temperatures and improve the level of thermal comfort. Taylor and Luther [21], Soudani et al. [22] and Fernandez et al. [23] believe that earthen materials, especially the rammed earth, are able to store large amounts of thermal energy and display latent heat owing to the liquid-to-gas phase change within the pores. However, as early as the creation of Yueling in the Book of Rites [24], there were records stating "in the month of Meng Chun, the weather falls, the earth's atmosphere rises, the heaven and the earth are in harmony, and vegetation sprouts," which accurately explains the changing law of the Earth's temperature field in spring. In addition, a great deal of importance has always been attached to agriculture in China. As early as in the Western Han Dynasty's agricultural work—the Book of "Fan Sheng Zhi Shu" [25]—the Chinese people have demonstrated a profound understanding of the relationship between seasons and soil, in order to guide agriculture production. The changing characteristics of the temperature fields and how they are affected by the seasons of spring, summer, autumn, and winter are explained in the book "Tian Ji Su Shu" [26]. All these quotes from literature reveal the existence and belief in the basic theory of relative latent heat.

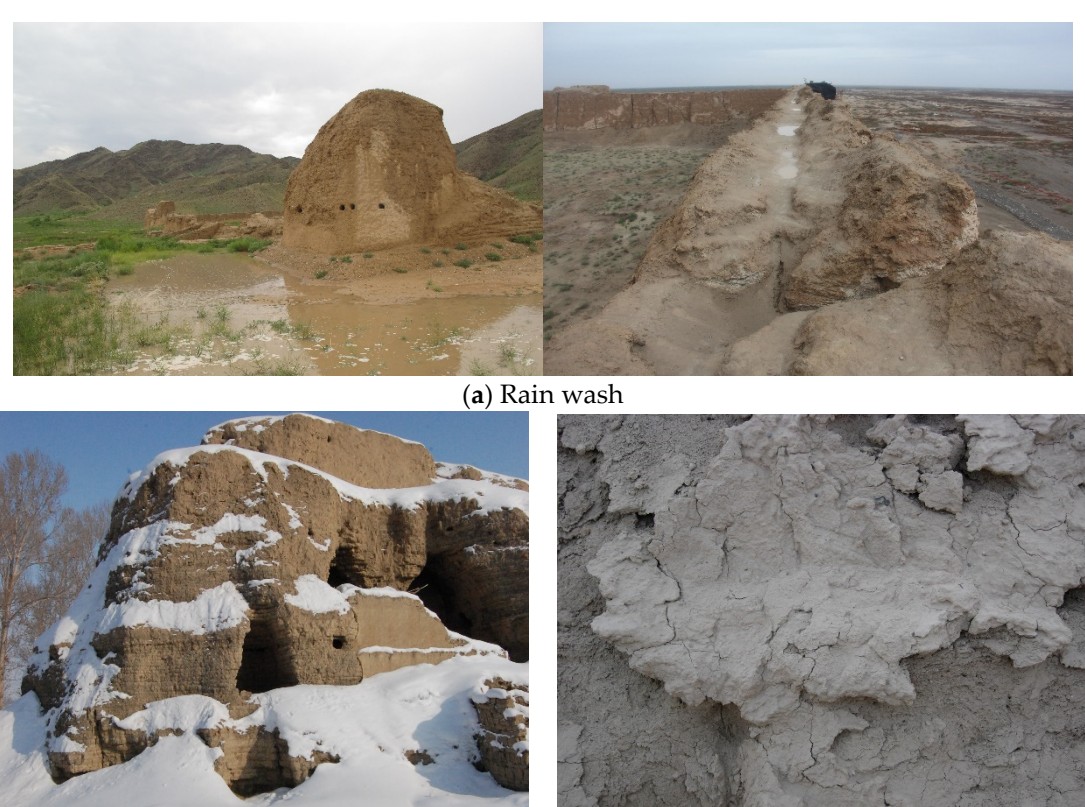

(**a**) Rain wash

(**b**) Site surface snow

(**c**) Weathered surface

**Figure 1.** Influence of natural conditions.

Extant research has noted that long-term dramatic temperature changes will cause expansion and contraction of the site body [27], as well as water and salt migration, water solidification [28,29], and volume expansion caused by water vapor [30]. In the process of repeated cyclic expansions and contractions, water and salt transport are enhanced (unsynchronized expansion and contraction of the earthen body, dissolution of soluble salts, crystalline expansion, re-shrinkage, and re-expansion), which leads to an obvious reduction in surface strength at the site and the appearance of multiple surface damages such as disruption, powdering, flaking, and peeling [31]. Among these causes, the sharp gradient changes in temperature fields are considered the internal driving force, and the changes to the solid, liquid, and gaseous phases of water are the main modes of energy storage and release (Figure 1).

In the past few years, increasing numbers of researchers and conservationists have begun to study the influence of temperature fields on sites. Some studies have concluded that strong temperature changes are one of the main causes of weathering on the surfaces of the earthen sites [32–34]. In particular, sharp rises and falls in the ambient temperature can rapidly increase or decrease the spacings between soil particles. Under repeated cycles of this process, the cohesion between soil particles decreases, soil surface becomes loose, and soil particles can even be loosened under external forces [28], the change of temperature field with severe gradient plays an important role [5]. Therefore, changes in the temperature fields with sharp gradients play an important role in the structural integrity at these sites.

However, owing to the influence of the temperature field, monitoring the temperature gradients of rammed earthen sites has always been a challenging problem. To solve this, it is necessary to further characterize the progressive gradient change processes of temperature fields, from the shallow surfaces to the insides of earthen sites. This study represents efforts toward the evaluation of the gradient temperatures of the surface and internal structure of a rammed earthen wall, which are monitored by embedding 5TM soil moisture temperature sensors and adopting traditional wall and atmospheric temperature testing

equipment. The test methods for the superficial layer and the internal temperature fields of large-volume rammed earth walls are also summarized in this paper. The variation characteristics and gradient laws of the 24 h wall temperature field of rammed earth sites in an arid environment were preliminarily characterized, including the gradual changes in the characteristics and evolution laws of the wall temperature fields in spring, summer, autumn, and winter. These variations were found to be extremely similar to the seasonal influence proposed in the ancient texts. This paper also proposed an empirical formula to describe the changes in the walls at rammed earthen sites with respect to atmospheric temperature. Further, the temperature field, stress field, and coupling relationships, as well as the weathering mechanisms that affect the surfaces of earthen ruins, were also investigated in this research.

## 2. Research Aim

The expansion and contraction stress caused by the gradient change of temperature field is one of the main factors leading to surface weathering of rammed earth sites. Therefore, the variation range and amplitude of the temperature gradient on the surface of rammed earth walls are the main indicators that lead to surface weathering. This paper discussed the influence of daily temperature difference and annual temperature difference on the walls through monitoring and revealed the sensitive areas and main seasons that affected the wall surface temperature, which provides a scientific basis for the prevention and control of surface weathering of rammed earth sites.

## 3. Methods

### 3.1. Test Wall Ramming

(1) Basic properties of test soil

The experimental soil was taken from platform soil, which is similar to the rammed soil of the Western Xia Mausoleum in Yinchuan, China, 80 km from No. 9 Western Xia Mausoleum. It belongs to loess, which is odorless and contains a small amount of flaky crushed stone. The mineral composition and particle distribution of the soil sample are given in Table 1.

**Table 1.** Basic properties of test soils.

| Mineral Components | | Distribution of Particle Size | | Basic Physical Properties | |
|---|---|---|---|---|---|
| Component | Content (%) | Particle Size (mm) | Content (%) | | |
| Quartz | 30 | 20–10 | 1.71 | Moisture content (%) | 1.84 |
| Calcite | 25 | 10–5 | 4.24 | Gravity (Gs) | 2.72 |
| Dolomite | 7 | 5–2 | 5.68 | Liquid limit (%) | 25.9 |
| Feldspar | 9 | 2–1 | 0.27 | Plastic limit (%) | 17.6 |
| Illite | 21 | 1–0.5 | 2.89 | Plastic index (IP) | 8.3 |
| Chlorite | 7 | 0.5–0.25 | 1.77 | Non-uniformity coefficient Cu | 8.27 |
| - | - | 0.25–0.075 | 28.63 | Curvature coefficient Cc | 1.17 |
| - | - | <0.075 | 64.66 | - | - |

(2) Rammed test wall

The test wall was rammed using traditional techniques [18]. The rammer had a mass of 6.9 kg with a diameter of 12 cm. The thickness of rammed pave soil is 12 cm, and each layer has been rammed six times for testing; the rammed test wall has a length of 3.4 m, the

height of 2 m, and bottom width of 1.05 m, with a north to south trend; after the ramming, the wall was trimmed to a length of 3.0 m, the height of 2.3 m, and bottom width of 1.1 m (Figure 2).

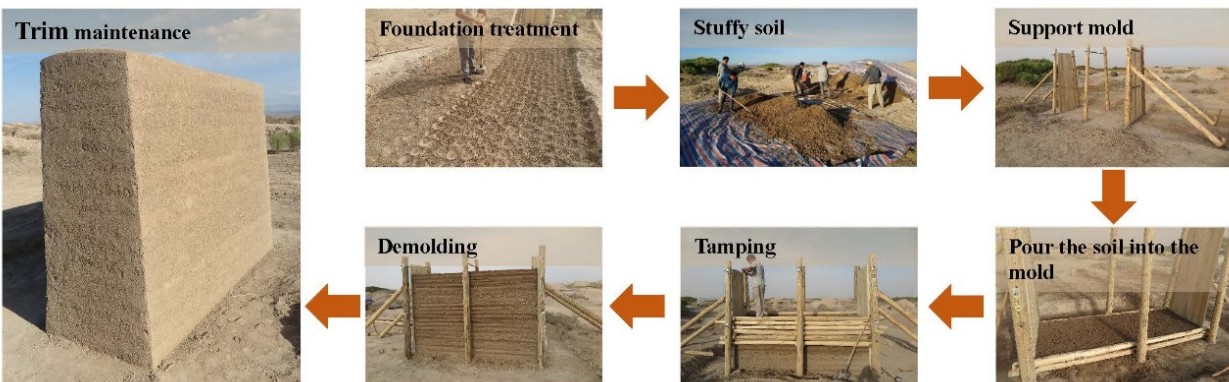

**Figure 2.** Testing wall tamping process.

*3.2. Monitoring Equipment and Installation of the Test Wall*

3.2.1. Monitoring Instruments

(1) 5TM temperature and humidity sensor

The 5TM sensor for soil temperature and humidity, produced by the Decagon Company (Lagos, Nigeria), was used for measurement. The sensor has the advantages of a small volume, strong corrosion-resistance, high measurement accuracy, and simple operation. The temperature measurement range was −40 to 60 °C, the humidity range was 0% to 100%, the measurement accuracy was 1 °C, and the measurement resolution was 0.1 °C. The temperature value of the soil was obtained by the thermistor on the sensor surface, and the temperature data measured by 5TM were collected by an EM50 collector (Figure 3).

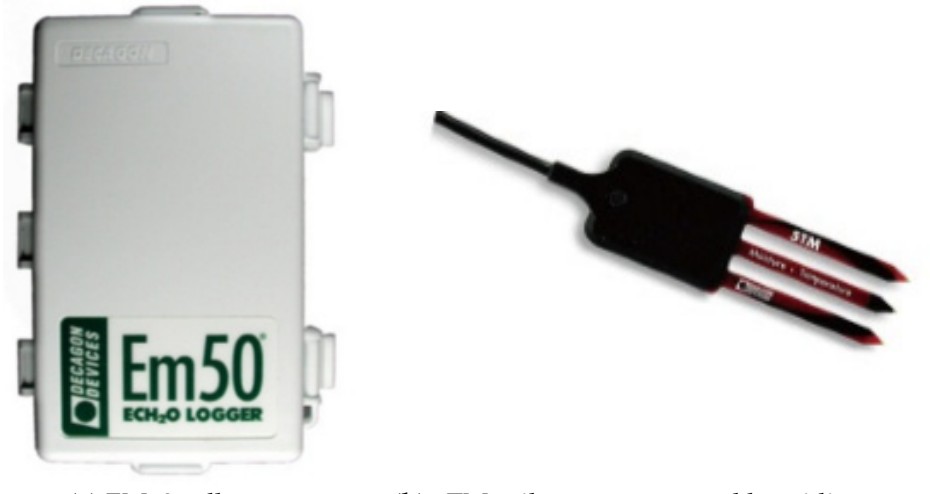

(**a**) EM50 collector　　　　　　　(**b**) 5TM soil temperature and humidity sensor

**Figure 3.** Testing instrument.

(2) Weather station

A meteorological station was set up at a distance of 10 m from the test wall to collect environmental data, such as ambient temperature, atmospheric relative humidity, atmospheric pressure, wind, rain, and dew, and data were sampled every five minutes. The collected data were compared and analyzed with data measured in the wall to provide basic information for analyzing the influence of the atmospheric environment on the temperature field of the wall (Figure 4).

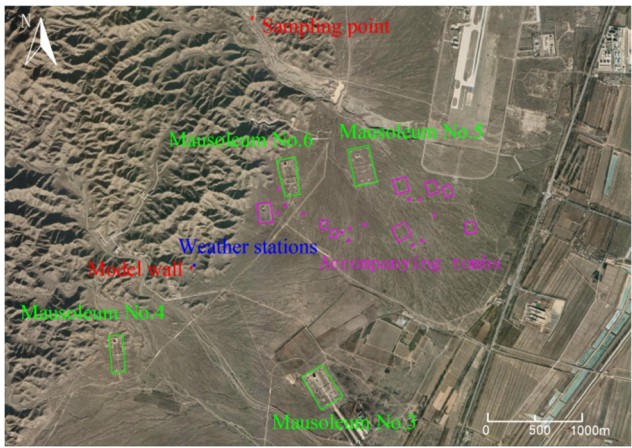

(**a**) Plane distribution map of the test area

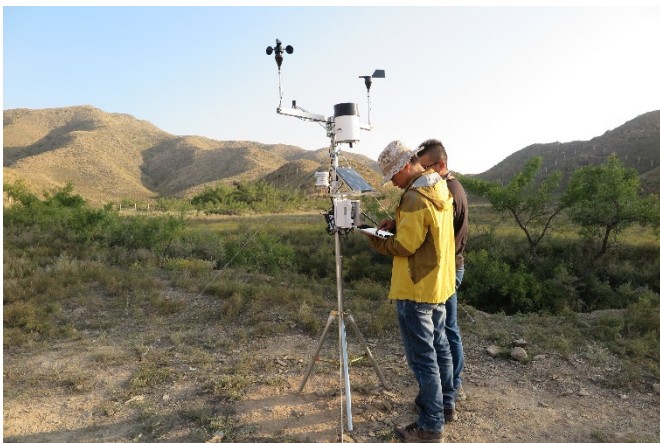

(**b**) Environmental data collection

**Figure 4.** Meteorological station.

### 3.2.2. Embedding Method for the Monitoring Equipment

(1) Sensor check

One end of the 5TM soil moisture temperature sensor was inserted into the collector and then connected with the computer through the matching data line. The time and date were corrected, and the parameters and numbers were then set according to the test requirements and instrument instructions.

(2) Sensor positioning

The density of the soil for the temperature field test was 1.70 g/cm$^3$, and the sensors were arranged inside the test wall in a total of 11 layers from bottom to top. Each layer, from A to D, was equipped with eight sensors that were symmetrically distributed around the central axis. Each layer from E to I was equipped with seven sensors that were symmetrically distributed around the central axis. The distance between the first sensor on both sides of the periphery of the site body and the surface of the test wall was 50 mm; the distance between the first sensor and the adjacent second sensor was at least 70 mm, and the rest of the sensors were all arranged at a distance of 100 mm. The sensors at the top and bottom of the test wall were arranged densely, the positions of the embedded sensors were marked, and the sensors were then placed on the marked points in sequence and recorded. The sensor arrangement is shown in Figure 5.

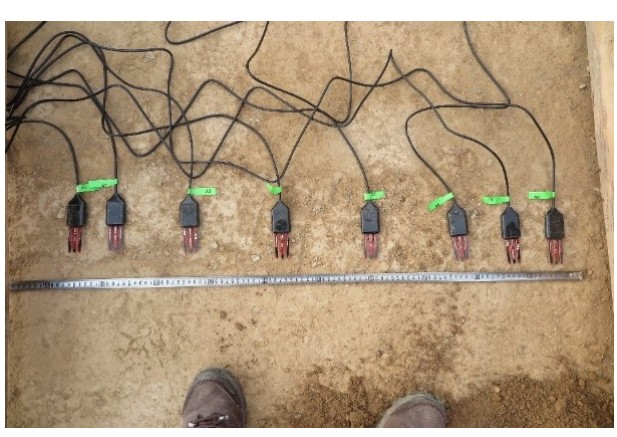

(**a**) 5TM sensor is laid out on site

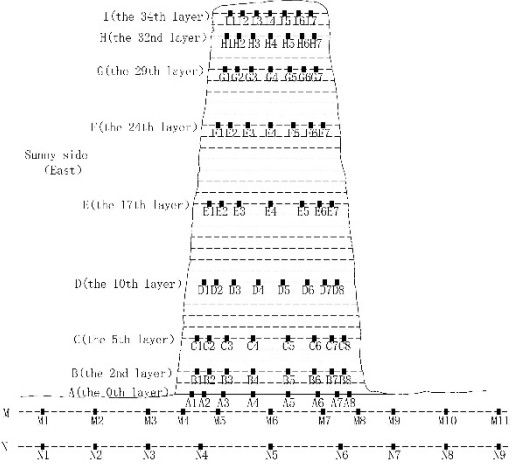

(**b**) 5TM sensor layout inside the wall

**Figure 5.** 5TM soil moisture temperature sensor layout.

(3) Sensor embedding

The virtual soil was paved and compacted under the sensor, to ensure that the sensor was in close contact with the soil. The sensor was fixed by hand, and the soil was then paved onto it to ensure that the sensor would not move when the virtual soil was paved.

(4) Ramming paved soil

The sensor leads were first fixed and then ordered. A small amount of experimental soil was then lightly pressed onto the leads to further fix them. When the sensor and leads were completely fixed, the soil was spread from the ground to the top surface of the test site body, from layer A to I. The test wall was divided into 34 layers; layer A was on the ground, there were two layers from layer A to layer B, three layers from layer B to layer C, five layers from layer C to layer D, and two layers between each layer from layer E to layer F, three layers from G to H, two layers from H to layer I, and one layer above I. The thickness of each layer was 59 mm. After the soil was paved, it was rammed six times with a rammer.

(5) Data acquisition and processing

Data were acquired every five minutes and processed to establish the position co-ordinate system of the test wall, divide the grid, and then draw a temperature isotherm diagram. The isotherms of the test wall at these two moments were drawn using a linear difference triangulation method in the Surfer software, in order to ensure the accuracy of the isotherm data.

## 4. Results

Owing to the periodic changes in the seasons of the year, the monitoring data of the temperature field of the test wall [17,19] were analyzed for one full year after the test wall was completely air-dried. In this study, the ambient temperature and test temperatures were measured at different positions on the test wall from 1 March 2017 to 1 March 2018, to comprehensively analyze how the wall was affected by the environment.

### 4.1. Seasonal Variation of Wall Temperature Field

Through continuous monitoring of the temperature field of the test wall, it was found that the lowest point and highest temperatures occurred at around 04:00 and 16:00 over one day, respectively. Figure 6 shows the temperature field isotherms at 04:00 and 16:00 for 12 consecutive months.

It can be seen from the isotherm at 04:00 in Figure 6 that the overall surface temperature is lower than the internal temperature of the test wall. With seasonal changes, the internal temperature was 1.5–8.5 °C higher than that of the surface. The maximum temperature difference (8.5 °C) occurred in April, whereas the minimum temperature difference (1.5 °C) occurred in August. The isotherms were distributed symmetrically along the central axis of the test wall. The ambient temperature was lower than the internal temperature of the test wall at 04:00 when there was no solar radiation. The test wall presented good uniformity in all directions and provided thermal insulation. The heat was uniformly dissipated from the test surface, and the overall surface temperature of the test wall was lower than its internal temperature.

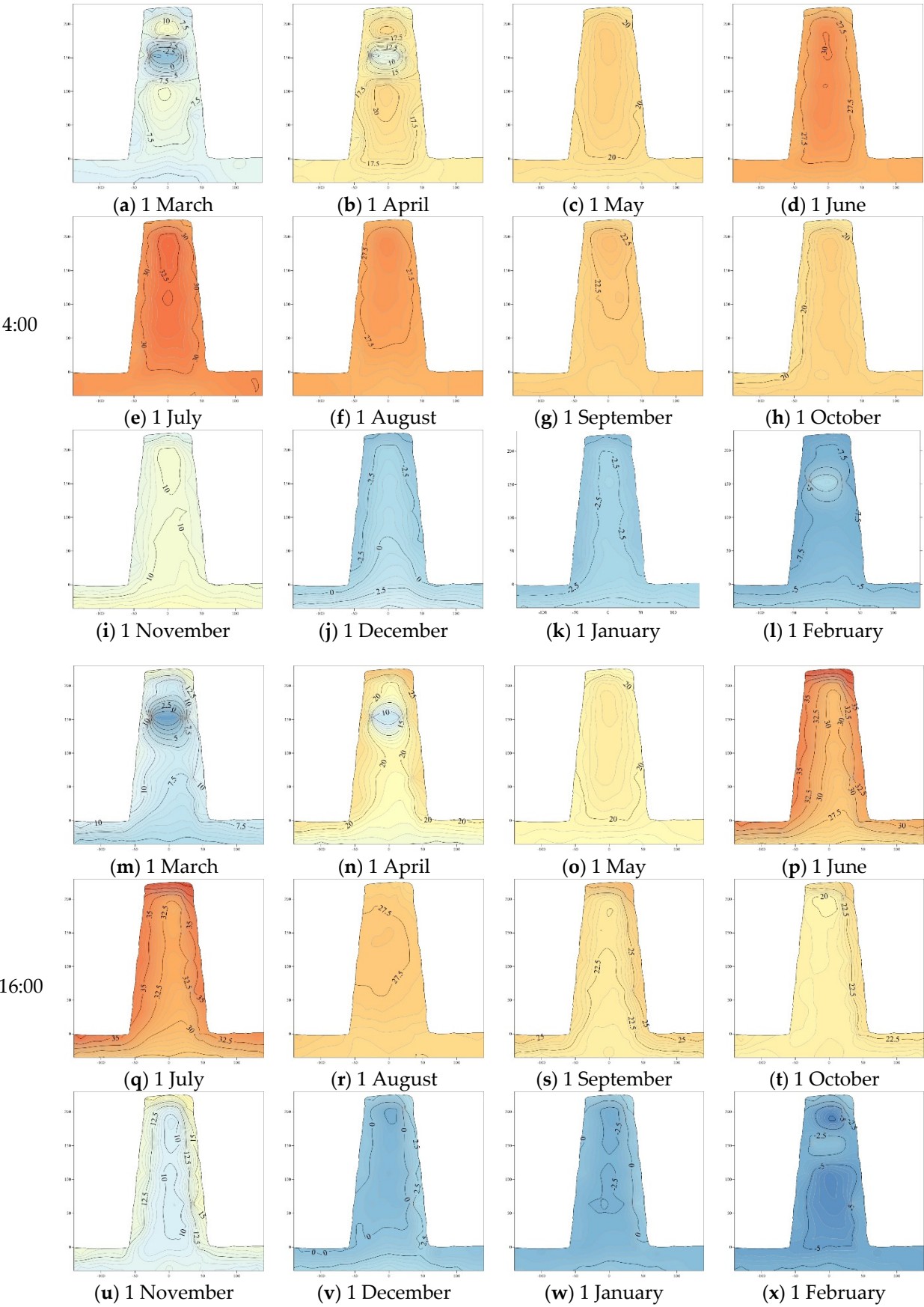

**Figure 6.** Trend charts showing changes in temperature field isotherms at 4:00 and 16:00 in one year.

In addition, as the foundation of the wall was affected by geothermal energy and the top was affected by solar radiation, the overall temperature of the wall changed relatively rapidly. However, the temperature in the range from 1.0 to 1.5 m from the middle-upper part of the wall was relatively constant. In January, the temperature of the entire wall was below zero, and the temperature decreased gradually from inside to outside, displaying a parabolic profile. In February, the temperatures at the top and foundation of the wall were higher than in the middle. In March, the temperatures of the wall were in the following order from high to low: top, bottom, and middle. In April, this changed to the bottom, top, and middle, and from May to September, the order changed to the middle, top, and bottom. The bottom temperature gradually decreased, and the area of the middle-temperature circle gradually shrunk. From October to December, the bottom temperature gradually rose, and the wall temperatures were in the following order from high to low: middle, bottom, and top.

The isotherm at 16:00 shown in Figure 6 reveals that the distributions of the internal and surface isotherms of the test wall are opposite to those at 4:00. With seasonal changes, the wall surface temperature was 1.5–12.5 °C higher than that inside. The maximum temperature difference was 12.5 °C (in July), and the minimum temperature difference was 1.5 °C (in August). The isotherms were asymmetrically distributed, and the top temperature was higher than that of the lower area of the test wall. With the change in the azimuth angle of solar radiation, at 16:00, the west facade of the wall was directly affected by the perpendicular incidence of the sun rays, and the temperature on the western surface increased rapidly. This asymmetric radiation caused the temperature of the west facade to be lower than that of the east facade; as the top of the test wall was exposed to direct solar radiation from the beginning to the end, its temperature was the highest. This was particularly true in July when the top temperature of the test wall could reach 46.99 °C.

### 4.2. Seasonal Diurnal Variation Characteristics of Wall Temperature Field

According to the changing trend of the temperature field isotherms in the above year, the temperature of the shallow surface layer of the test wall changed with the changes in the ambient temperature, and when the ambient temperature changed rapidly, the temperature of the shallow surface layer of the test wall also changed rapidly. The change in ambient temperature had little influence on the temperature change inside the test wall, and it decreased stepwise from the outside to the inside. The area where the wall temperature changed by more than 10 °C across the four seasons was termed the sensitive area; the area where the wall temperature changed between 5 and 10 °C was termed the transition area; the area where the wall temperature changed between 1 and 5 °C was termed as the stable area, and the area where the wall temperature changed by less than 1 °C was termed as the constant area. Considering the climate change in the four seasons, efforts were made to reduce the influence of extreme weather changes such as clouds, rainfall, wind, and snow on the solar radiation received by the test wall. To this end, monitoring data of the representative spring equinox, summer solstice, autumn equinox, and the best sunshine days near the winter solstice in the four seasons were collected for comparative analysis.

#### 4.2.1. Spring

As seen in Figure 7, sunshine was the strongest from 00:00 on 22 March to 00:00 on 23 March 2017. Starting from 00:00 on 22 March, the data were collected every 4 h to complete the isotherm diagrams, as shown in Figure 8.

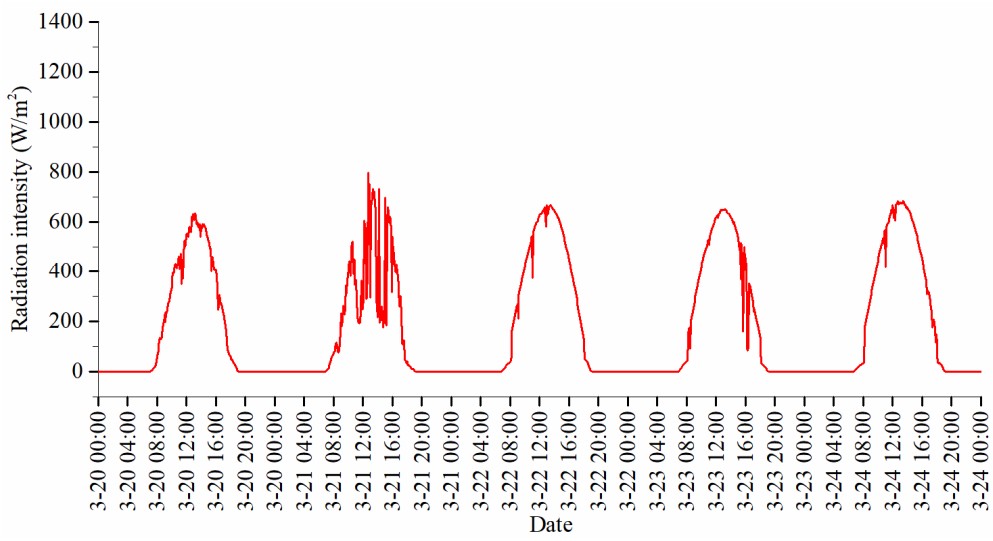

**Figure 7.** Selection of optimum days for spring illumination.

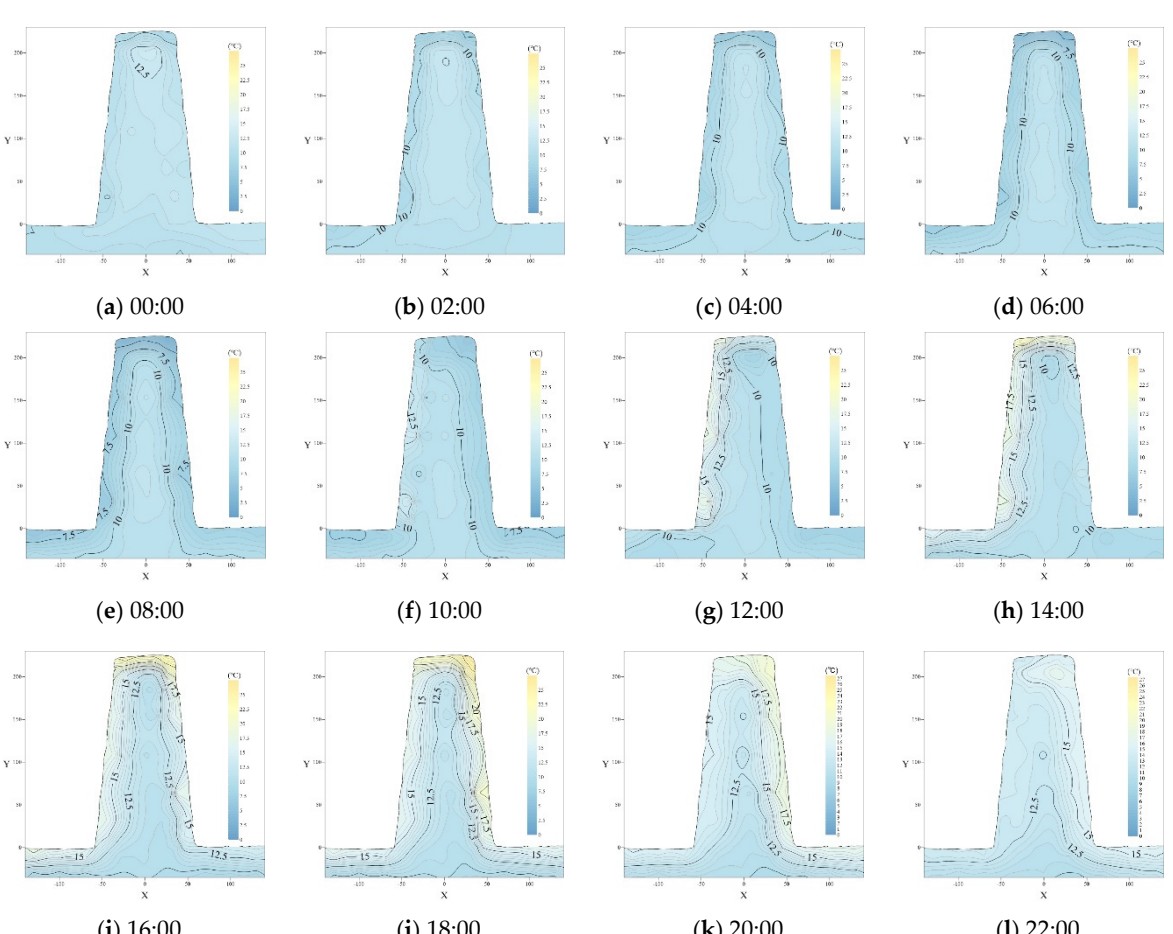

**Figure 8.** Temperature variations in the test wall on 22 March.

As can be seen from Figure 8, the temperature inside the test wall is almost constant for one day in spring (March), whereas the surface temperature changed greatly. The temperature field change of the wall surface can be divided into three stages: the temperature gradually fell down to the lowest from 00:00 to 08:00, rose up to the highest from 08:00 to 16:00, and dropped slowly from 16:00 to 24:00. These observations show that the wall temperature decreased from 16:00 to 08:00 on the next day. To be specific, it decreased

slowly from 16:00 to 20:00, and then decreased rapidly from 20:00 to 08:00 on the next day. From 08:00 to 14:00, the temperature of the wall gradually increased, and the highest temperature in this stage occurred in the top corner area on the east side of the test wall, reaching 17.5 °C. The wall temperature was the highest between 14:00 and 16:00, and the hottest area of the test wall gradually moved from the east to the west, i.e., to the corner area of the west façade; and the maximum temperature was 26 °C.

According to the above changes in the different temperature fields, the wall was divided into four different areas: the sensitive area, transition area, stable area, and constant area, among which the sensitive area was divided into the strong sensitive area and sensitive area. The strong sensitive area is the red-filled part on the west side of the top of the wall in  Figure 9, and it shows the largest temperature change. The wall temperature at 16:00 was 20 °C higher than the temperature at 08:00; this area was located 16.17 cm from the western side and 20.81 cm from the top of the wall. The sensitive area is the orange-colored part in Figure 9, which occupies the top of the wall and the surface layer on the upper half of the west side of the wall. The thickness of this top layer was 14.55 cm, its thickness on the west side was 10.20 cm, and its height was 82.24 cm. The temperature at 16:00 was 12.5 °C higher than that at 08:00. The transition area is the green-filled part in Figure 9. Compared to that at 08:00, the temperature rose by 5 to 8 °C. The thickness of this area was about 20 cm, with the thicknesses of 21.28 cm on the east side, 12.26 cm on the top, and 22.85 cm on the west side. The stable area is colored cyan (Figure 9). Compared with the temperature at 8:00, the temperature rose by 1–3 °C; the east side was 42.11 cm away from the wall surface, and the west side was 25.26 cm away from the wall surface. The constant region is the blue-filled part (Figure 9), having a roughly triangular shape. The height of the constant region was 102.96 cm, and the temperature remained almost unchanged within this area.

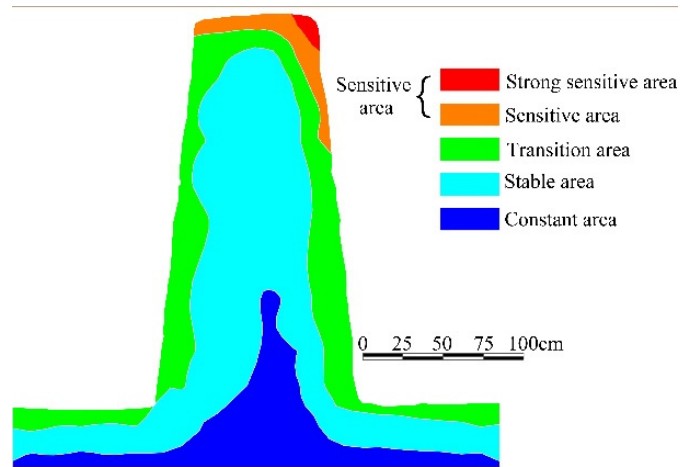

**Figure 9.** Temperature field of test wall in spring.

To summarize, the spring temperature field was divided into four areas, as shown in Figure 9 and the average temperature change in the sensitive area was above 10 °C. The temperature change in the transition area was 5–8 °C, and that in the stable area was 1–3 °C; the temperature in the constant area remained unchanged compared, as opposed to the other time periods (Figure 9).

4.2.2. Summer

As can be seen from Figure 10, the sunshine was brightest from 00:00 on 16 June to 00:00 on 17 June 2017. Starting from 00:00, data were collected every 4 h to complete the isotherm chart, as shown in Figure 11.

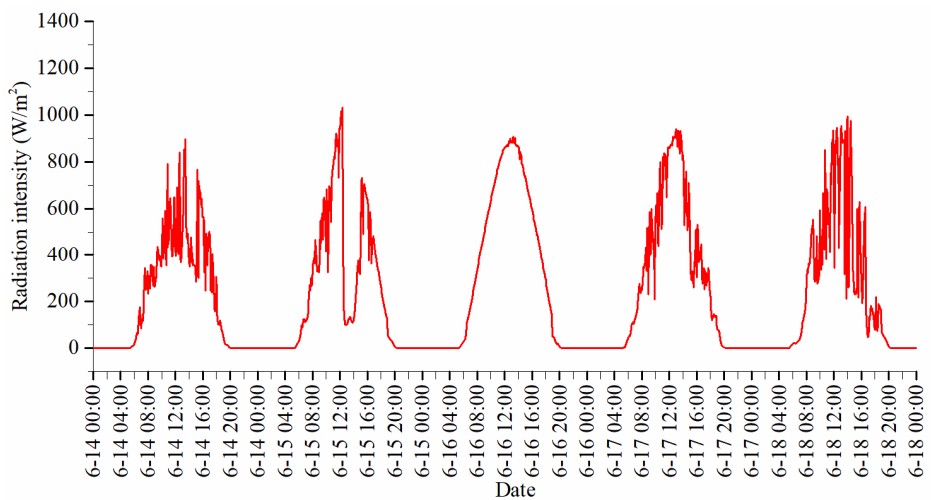

**Figure 10.** Selection of optimum days for summer illumination.

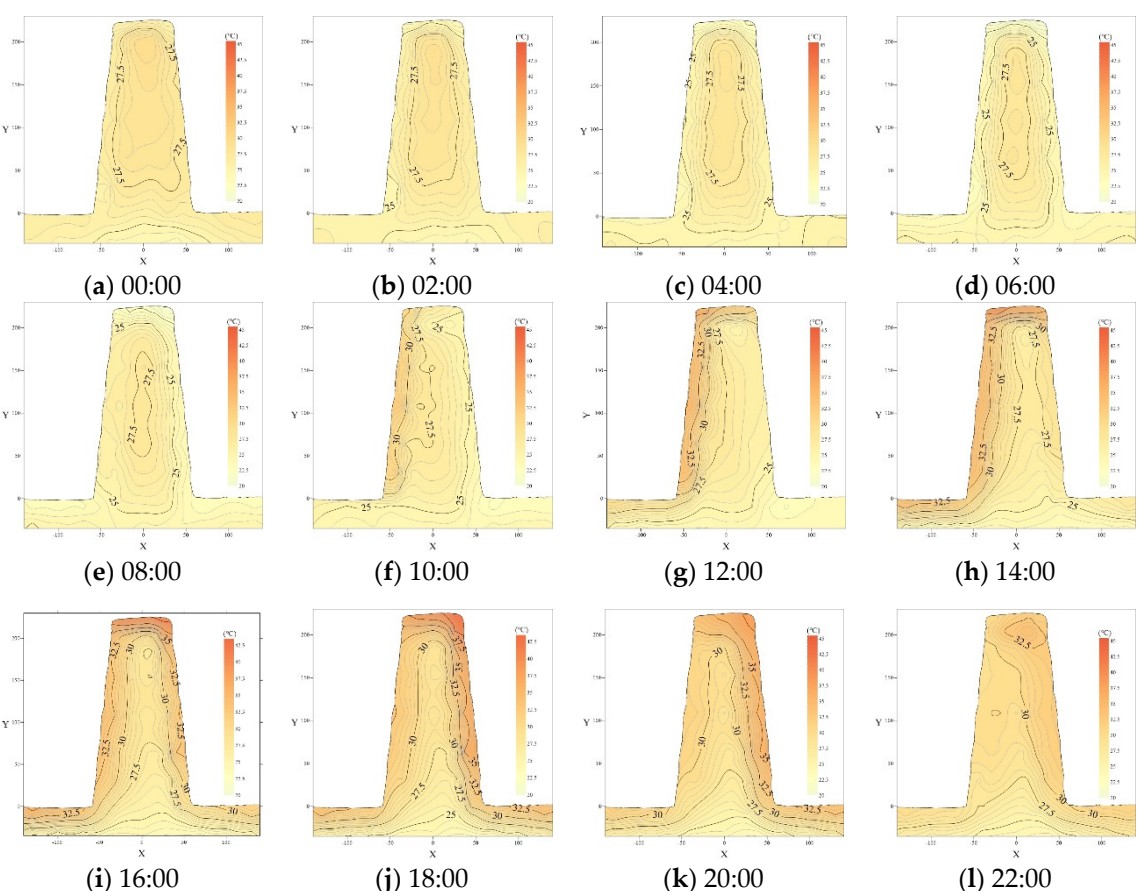

**Figure 11.** Temperature variation of test wall on June 16.

As can be seen from Figure 11, the internal temperature of the test wall was almost constant during this day in summer, and the largest temperature change was observed on the wall surface. The temperature gradually fell down to the lowest from 00:00 to 08:00, rose up to the highest from 08:00 to 16:00, and dropped slowly from 16:00 to 24:00. This indicates that the wall temperature gradually rose from 08:00 to 14:00, and the highest temperature was observed in the top corner area to the east of the test wall; the maximum temperature was 37.5 °C in this stage. The wall temperature was highest from 14:00 to 16:00, and the highest temperature area moved gradually from the east to the west, i.e., to

the corner area on the west façade of the test wall, with a maximum temperature of 40 °C. The wall temperature decreased from 16:00 to 08:00 on the next day. It decreased slowly from 16:00 to 20:00 and then decreased sharply from 20:00 to 08:00 on the next day. The temperature curve gradually changed from an asymmetric to a symmetric elliptical spiral, and the surface temperature dropped to 22 °C.

According to the change in the summer temperature field, the wall was divided into four regions, as shown in Figure 12. The strong sensitive area is the red-filled part in the upper right corner at the top of the wall; this area exhibited the largest temperature change. Compared with the value at 08:00, the temperature had increased by 11.5 °C at 16:00. This area had a width of 25.54 cm and a height of 27.01 cm at the top west side of the wall. The sensitive area is the orange-filled area in Figure 12. It was distributed 15.58 cm from the top to the inside of the wall and 17.24 cm from the surface to the inside of the west side of the wall. Compared with the value at 08:00, the temperature had increased by 11.5 to 12.5 °C at 16:00. The transition area is the green-filled part in Figure 12. It was distributed around the wall as a parabolic shape, with a thickness of 15 cm. The maximum distance from the east side to the wall surface was 16.04 cm, the thickness at the top was 17.78 cm, and the distance from the foundation was 18.06 cm. Compared with the value at 08:00, the temperature had increased by 6 to 10 °C at 16:00. The stable area is the cyan-colored area in Figure 12. It was also distributed parabolically around the wall, with a thickness of 12.36 cm on the top, 19.66 cm on the east side, and 11.90 cm on the west side. Compared with the value at 08:00, the temperature increased by 4 °C at 16:00. The constant area is the blue-filled part; it exhibited almost no change in temperature, and its area increased by 2.65 times compared to that during spring.

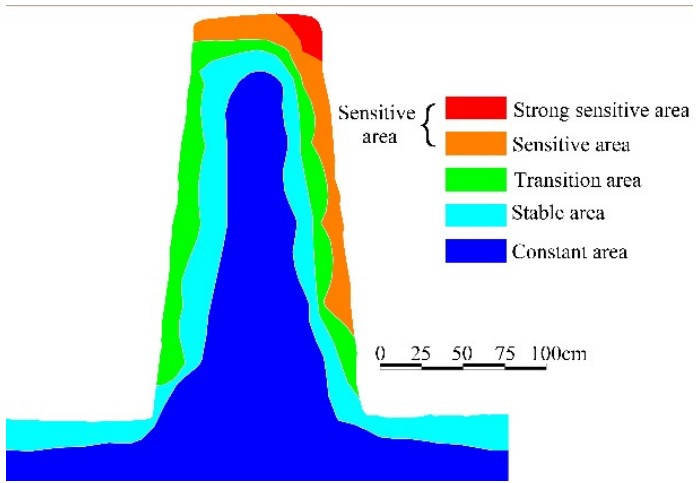

**Figure 12.** Temperature field of test wall in summer.

In summary, the summer temperature field was divided into four areas: the average temperature change in the sensitive area was above 10 °C; the temperature change in the transition area was 5–8 °C, and the stable area was 1–3 °C, which remained unchanged compared with the other time periods in the constant area, as seen in Figure 12.

### 4.2.3. Autumn

As can be seen from Figure 13, the sunshine was strongest from 00:00 on 22 September to 00:00 on 23 September. Starting from 00:00, data were collected every 4 h to complete the isotherm diagram, as shown in Figure 14.

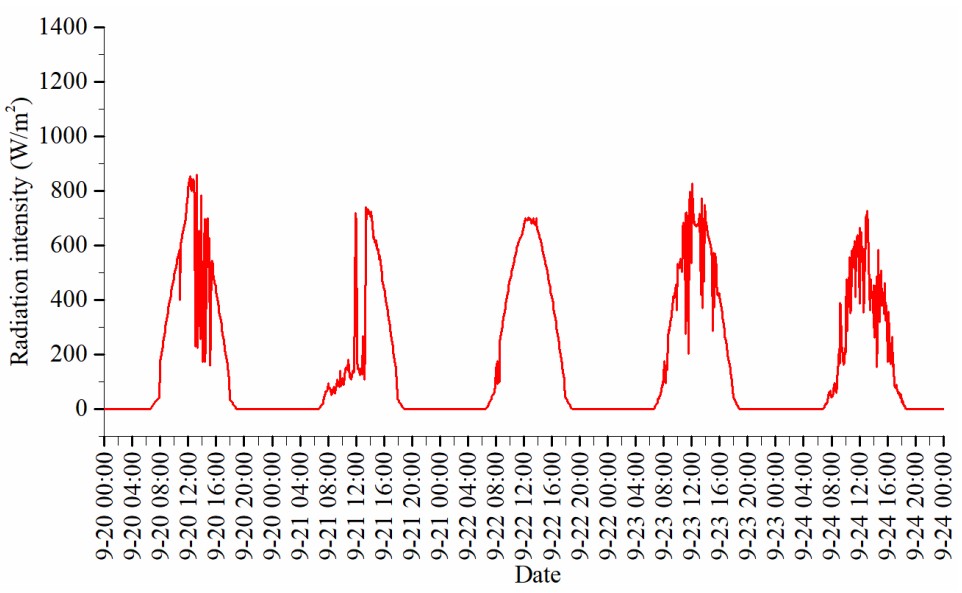

**Figure 13.** Selection of optimum days for autumn illumination.

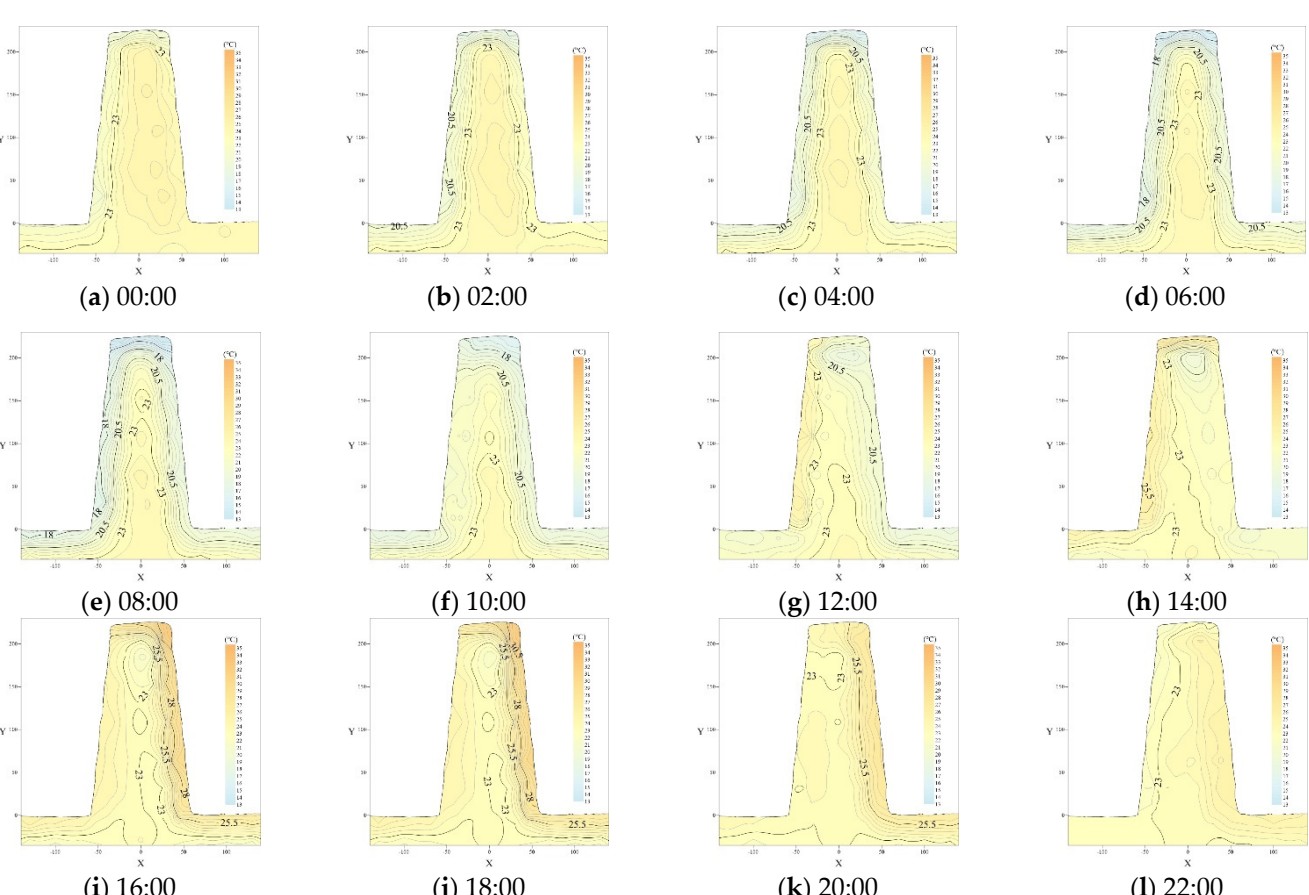

**Figure 14.** Temperature variations in the test wall on 22 September.

As can be seen from Figure 14, the internal temperature of the test wall was almost constant across one day in autumn, and the surface temperature changed the most. The temperature gradually fell down to the lowest from 00:00 to 08:00, rose up to the highest from 08:00 to 16:00, and dropped slowly from 16:00 to 24:00. This indicates that the wall temperature gradually rose from 08:00 to 14:00, and the highest temperature was observed in the top corner area to the east of the test wall; and the maximum temperature was 28 °C

in this stage. The wall temperature was the highest from 14:00 to 16:00, and the highest temperature area moved gradually from the east to the corner area on the west facade of the test wall; it reached 40 °C. The wall temperature decreased from 16:00 to 08:00 on the next day; it decreased slowly before decreasing rapidly from 20:00 to 08:00. The decreasing trend gradually changed from an asymmetrical to a symmetrical parabola, and the surface temperature dropped to 16 °C.

According to the changes in the autumn temperature field, the wall was divided into four regions, as shown in Figure 15. The strong sensitive area is the red-filled part, with the largest temperature change. The wall temperature at 16:00 was 17 °C higher than at 08:00. This area was located 17.28 cm from the west wall and 34.07 cm from the top of the wall. The sensitive area is the red-filled part, which was located 32.14 cm away from the top and west edges of the wall. The maximum depth was 20.82 cm from the surface to the inside of the west wall, and it extended up to 16.09 cm away from the bottom of the wall. The wall temperature at 16:00 was 10–17 °C higher than at 08:00. The transition area is the green-filled part; it was distributed on the top and west of the wall. The thickness of the top layer was 15.66 cm, the west side was 17.26 cm thick, and the depth was 12.41 cm. The wall temperature at 16:00 was 10–17 °C higher than at 08:00. The constant area is the cyan-filled part. It was roughly triangular. The height of the constant region was 228.93 cm, its width was 44.75 cm, and height was slightly larger than that of the constant region in summer. The width was 1.2 times that of the constant region in summer, and the temperature remained almost unchanged in this area.

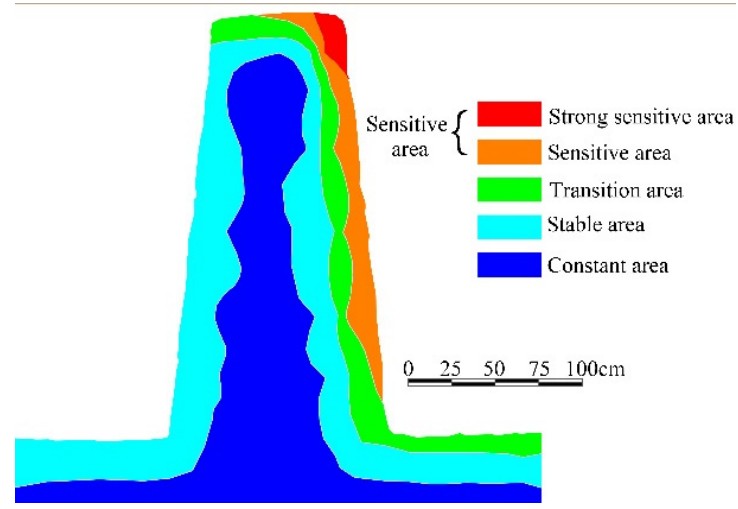

**Figure 15.** Temperature field of test wall in autumn.

To summarize, the autumn temperature field was divided into four areas, as shown in Figure 15; the average temperature change in the sensitive area was above 10 °C. The temperature change in the transition zone was between 5 and 8 °C, and the temperature change in the stable region was 1–3 °C. The temperature in the constant area did not change, compared with the other time periods in the constant region in Figure 15.

### 4.2.4. Winter

As can be seen from Figure 16, the sunshine was the brightest and most representative from 00:00 on 24 December to 00:00 on 25 December.

According to Figure 17, the internal temperature of the test wall is below zero. The internal temperature was almost constant, and the surface temperature did not change much across one day in winter. The temperature gradually fell down to the lowest from 00:00 to 08:00, rose up to the highest from 08:00 to 16:00, and dropped slowly from 16:00 to 24:00. The wall temperature gradually rose from 8:00 to 14:00, and the highest temperature was observed at the top corner area of the east side of the test wall (3 °C). The wall

temperature was the highest from 14:00 to 16:00, and the highest temperature area was from the east to the west of the corner area on the west façade of the test wall (5 °C). The wall temperature decreased from 16:00 to 08:00 on the next day; it decreased slowly from 16:00 to 20:00, falling by 0.7–3 °C and then decreased rapidly from 20:00 to 08:00 on the next day. The curve gradually changed from a symmetric elliptic spiral to an asymmetric shape, and the surface temperature uniformly decreased to 6 °C.

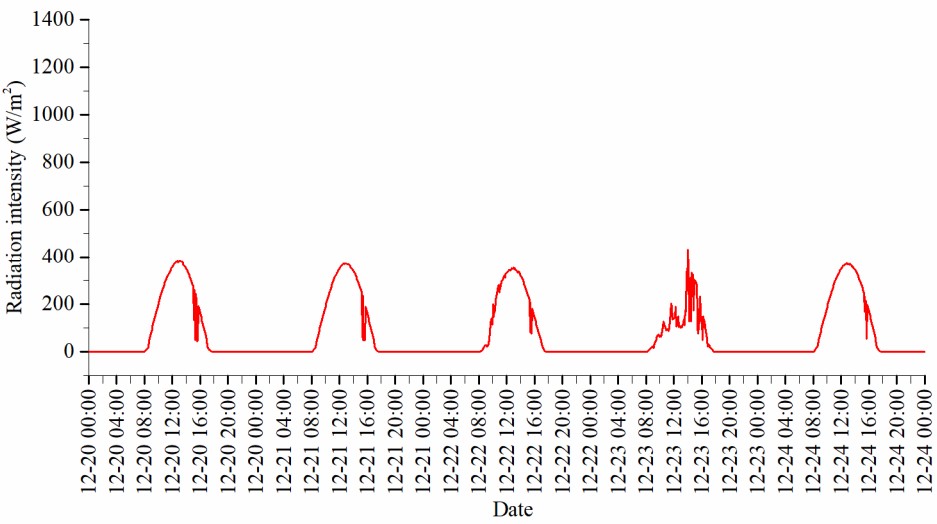

**Figure 16.** Selection of optimum days for winter illumination.

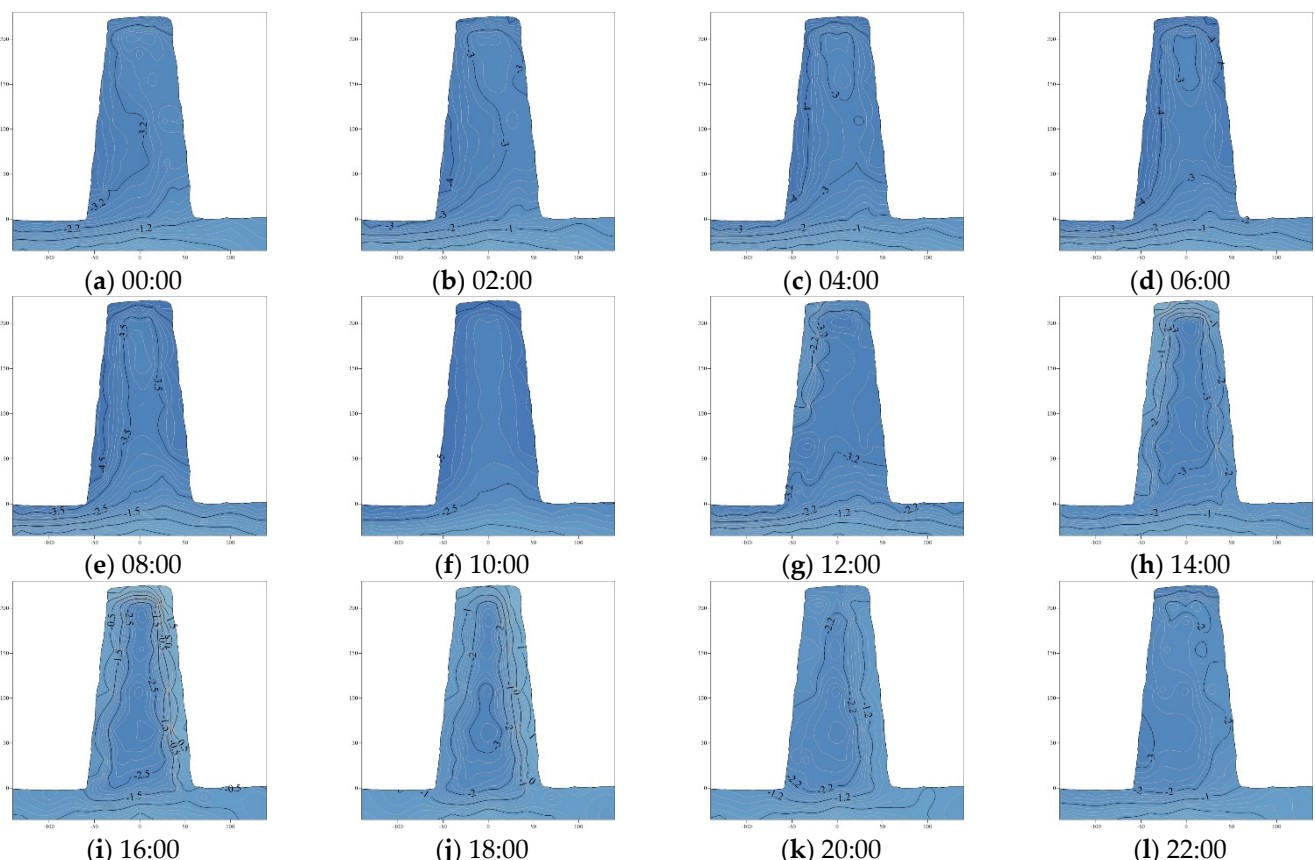

**Figure 17.** Temperature variations in the test wall on 22 December.

As can be seen from Figure 18, the overall temperature difference of the wall is less than 10 °C, so there was no sensitive area, only a transition area, a stable area and a constant area. The transition area is the green-filled part, which showed the largest temperature change. The wall temperature at 16:00 was −6 °C higher than it was at 08:00. The area was located 37.93 cm away from the west edge and 51.37 cm from the top of the wall. The constant area is the cyan-filled part in the middle and upper part on the east, west, and top of the wall. It was 13.66 cm thick on the top of the wall, 22.22 cm thick on the east side, and 107.28 cm deep from the foundation of the wall. The west side was at a depth of 20.18 cm. The constant area is the blue-filled part, which occupied 80.64% of the cross-sectional area of the entire wall. The temperature was almost unchanged in this range.

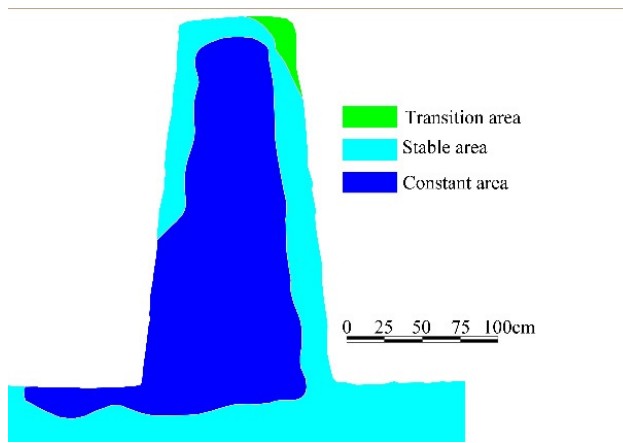

**Figure 18.** Temperature field of test wall in winter.

To summarize, the winter temperature field was divided into three areas (Figure 18), and the average temperature change in the transition area was 6 °C. The temperature change in the stable region was 2–3 °C. The third area, namely the constant area, showed no change in temperature compared with the other time periods Figure 18.

### 4.3. Variations in the Wall Temperature Field at Different Positions

The trend of the test wall temperatures and changes in the solar radiation angle meant that the test wall experienced different temperature variations during different seasons, at their different positions and depths, within one year. The temperature changes on the east and west sides of the wall within one year are shown in Figure 19. The temperature changes at the top, middle, lower, west, and east areas of the wall under the highest and lowest atmospheric temperatures are shown in Figure 20.

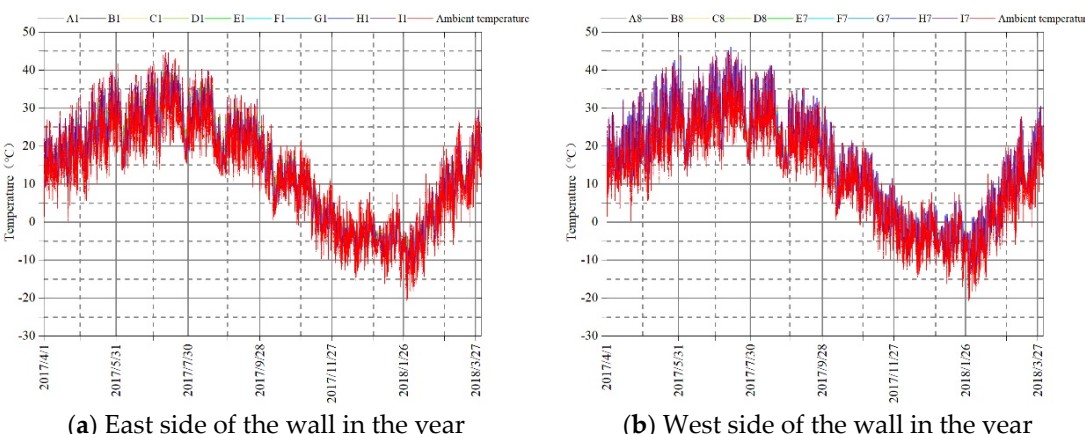

(**a**) East side of the wall in the year　　　　　　(**b**) West side of the wall in the year

**Figure 19.** Temperature change characteristic on the wall in the year (1 April 2017–31 March 2018).

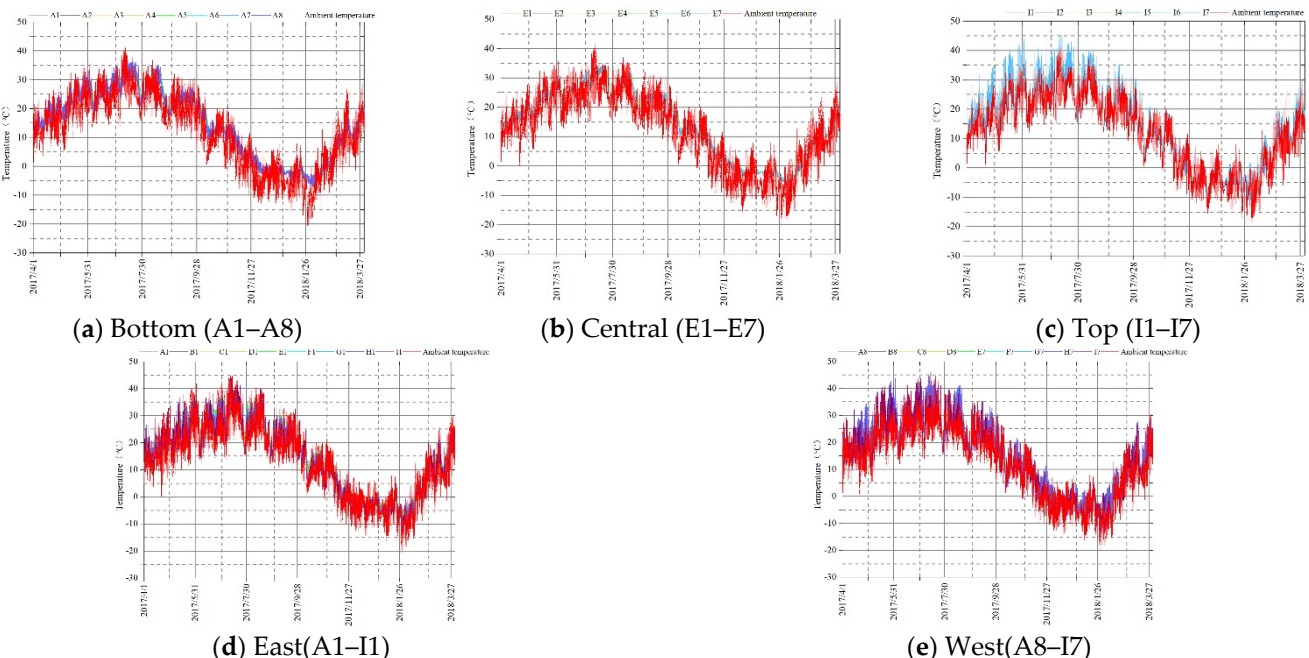

**Figure 20.** Wall temperature-atmospheric temperature relationship (1 April 2017–31 March 2018).

As can be seen from Figure 19, the temperature changes on the east and west sides of the wall were the same as those of the atmosphere, but the surface temperature of the wall changed rapidly. Within one year, the maximum temperature of the atmospheric environment was 41.24 °C, whereas the maximum temperature on the west side of the wall was 46.99 °C. The maximum temperature on the east side of the wall was 44.20 °C, which was 2.96 °C higher than the atmospheric temperature. The lowest atmospheric temperature was −20.5 °C. The lowest temperature on the west side of the wall was −15.2 °C, which was 5.3 °C higher than the atmospheric temperature, the lowest temperature on the east side of the wall was −15.4 °C, which was 5.1 °C higher than the atmospheric temperature. The maximum atmospheric temperature difference was 61.74 °C within one year, and the maximum temperature difference of the wall was 62.39 °C. The temperature of the east wall was not obviously warmer than the atmospheric temperature, while the temperature of the west wall was almost always higher than the atmospheric temperature, which was directly related to solar radiation.

As can be seen from Figures 20–26, the changes in temperature at the bottom, middle, top, east, and west of the wall showed the same trends, the changes in temperature at the bottom, middle, top, east, and west of the wall showed the same trends as that of the atmospheric temperature. However, the overall trend had a certain lag, which was no more than 6 h. As can be seen from Figure 20, there were severe wall temperature changes (from strong to weak) in the top, west, east, bottom, and middle areas. Along the vertical direction of the wall, the sensitivity of temperature to environmental influences, from strong to weak, ranged from the top to the middle and then to the bottom. Within one year, the highest temperature at the top of the wall was 46.99 °C, the lowest temperature was −15.3 °C, and the temperature difference was 62.29 °C. The highest temperature at the bottom of the wall was 37.3 °C, the lowest temperature was −5.9 °C, and the maximum temperature difference was 43.2 °C. The highest temperature in the middle was 35.8 °C, the lowest temperature was −5.6 °C, and the maximum temperature difference was 41.4 °C. Along the horizontal direction of the wall, the temperature change was more severe in the west than in the east, and it was relatively stable in the middle. The annual maximum temperature difference on the west side was 62.6 °C, the maximum temperature difference on the east side was 60 °C, and the maximum temperature difference in the middle was 36.8 °C.

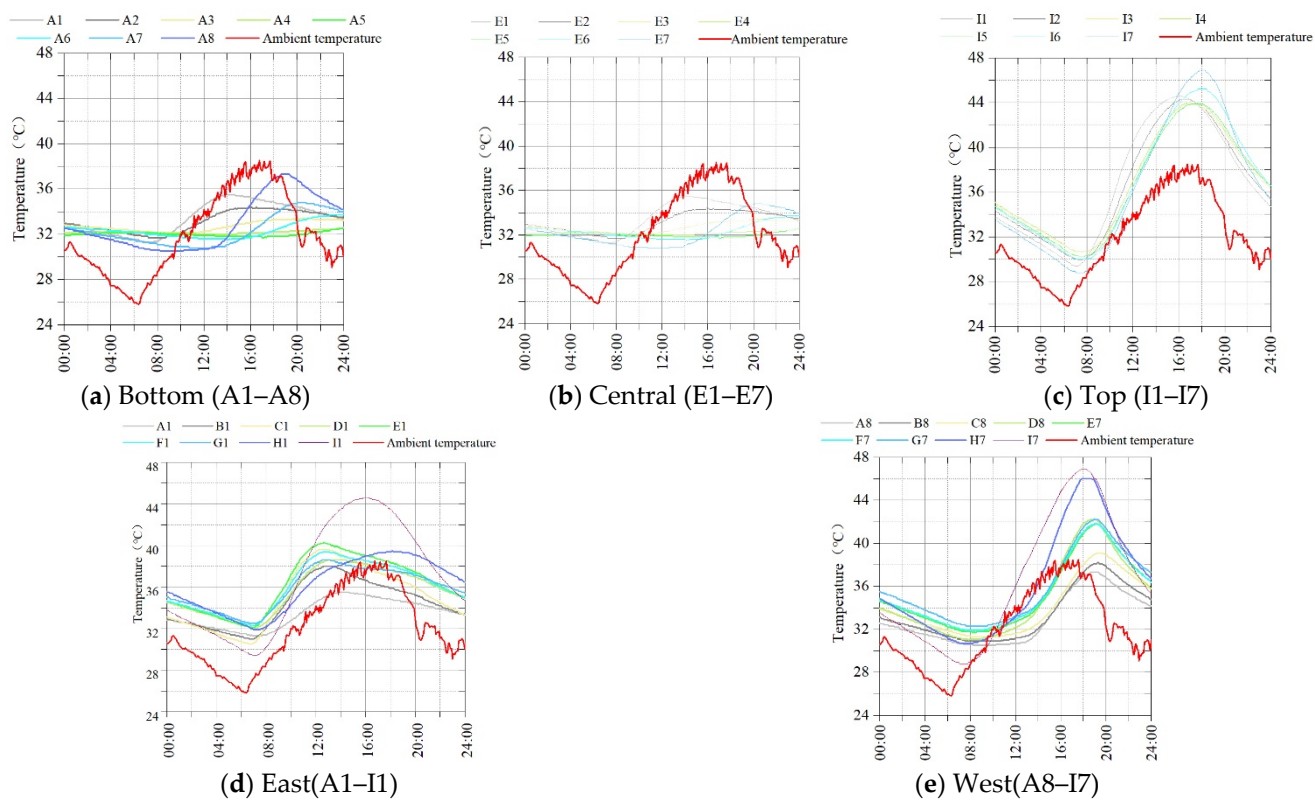

**Figure 21.** Variation of wall temperature during the highest atmospheric temperature throughout the year (13 July 2017).

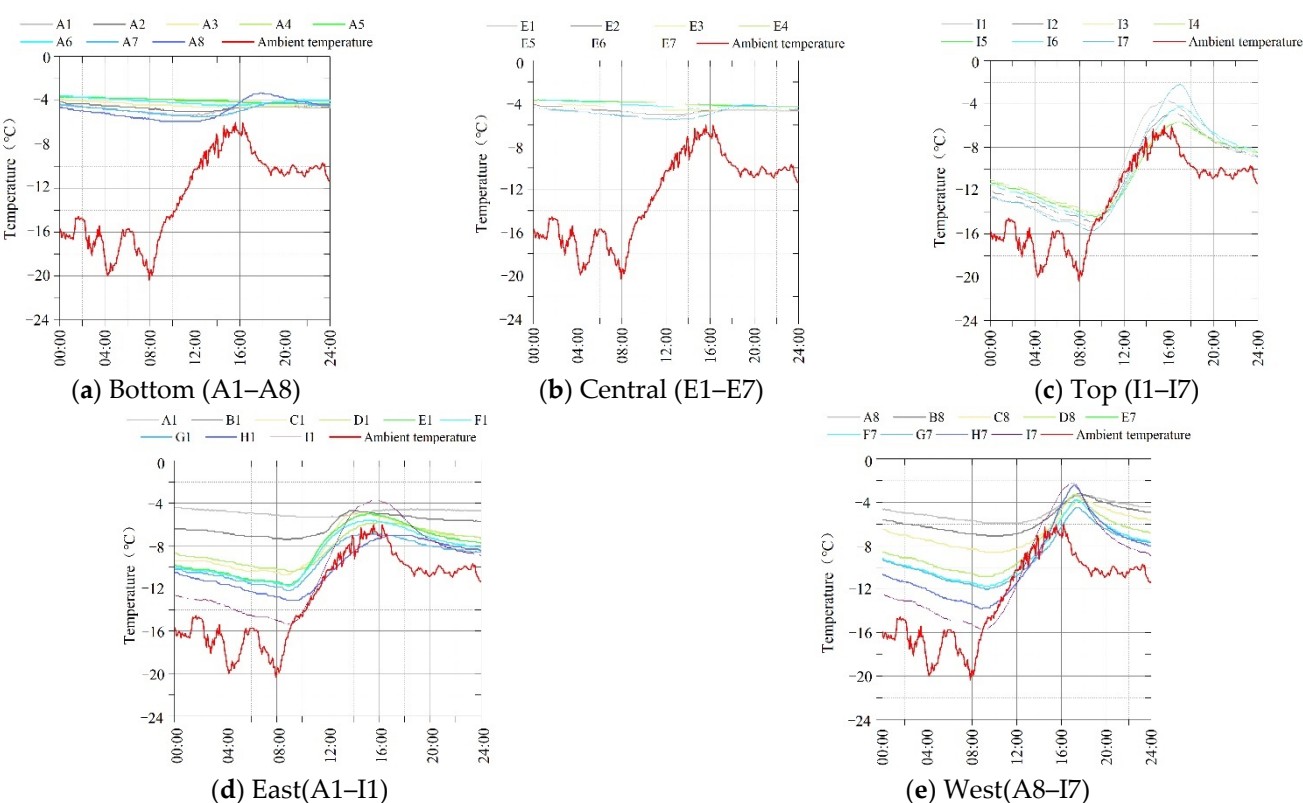

**Figure 22.** Variation of wall temperature during the lowest atmospheric temperature throughout the year (19 January 2018).

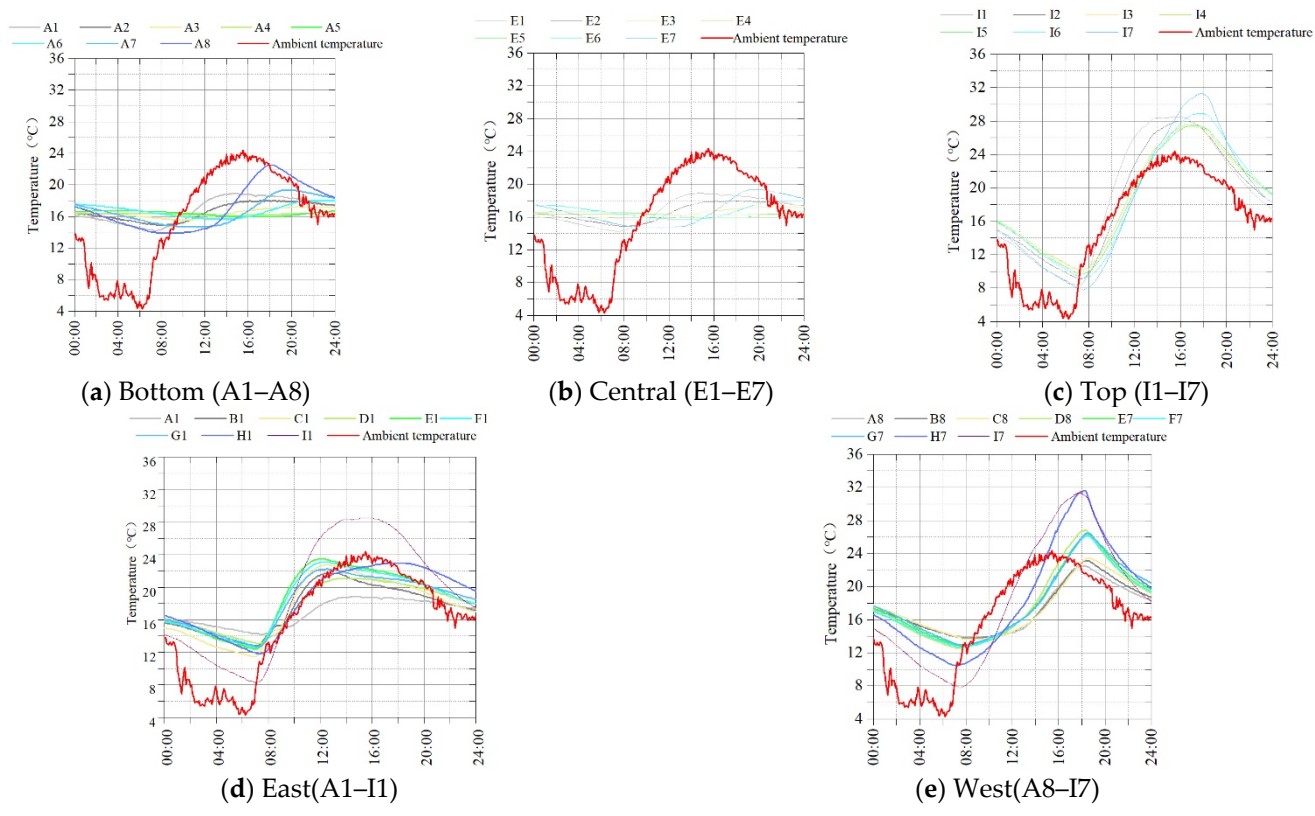

**Figure 23.** Diurnal wall temperature change with maximum atmospheric temperature difference in spring (22 April 2017).

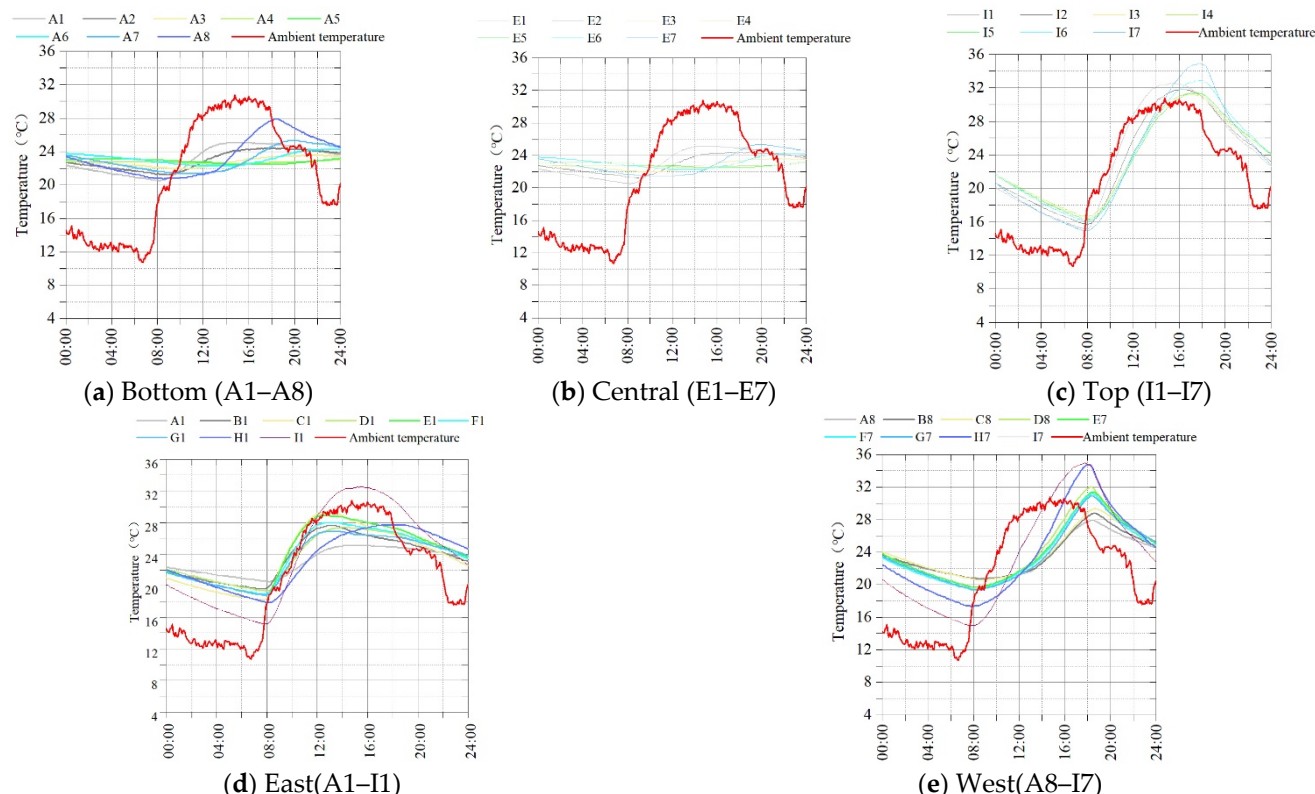

**Figure 24.** Diurnal wall temperature change with maximum atmospheric temperature difference in autumn (6 September 2017).

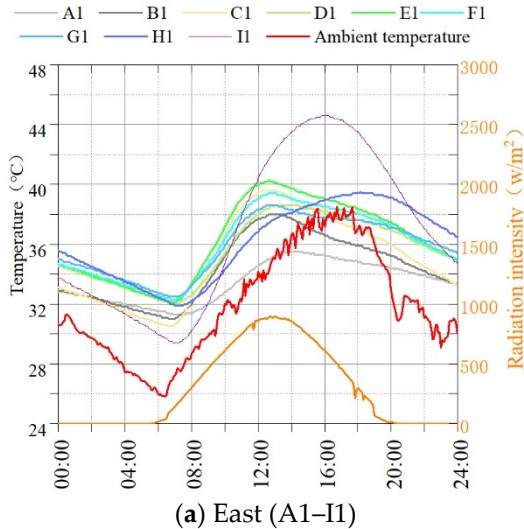
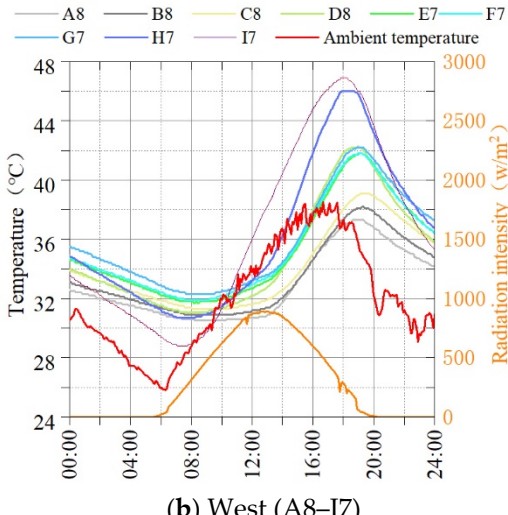

**Figure 25.** The relationship of wall temperature, atmospheric temperature and the strongest solar radiation (13 July 2017).

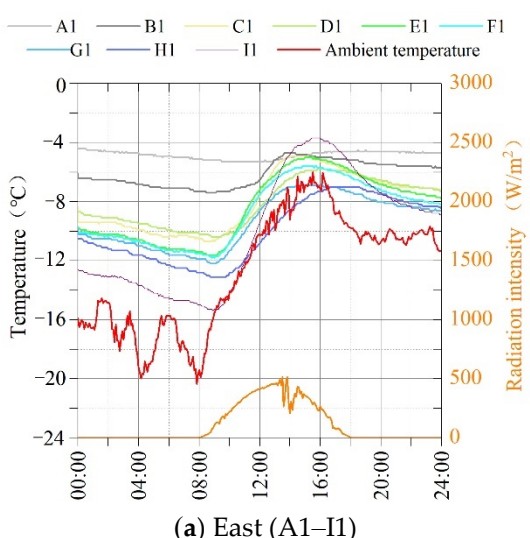
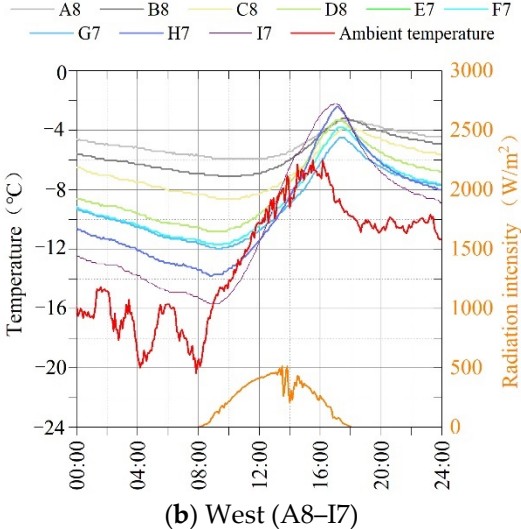

**Figure 26.** The relationship of wall temperature, atmospheric temperature and the weakest solar radiation (19 January 2018).

As can be seen from Figure 21, on the hottest day of the year (13 July 2019), the maximum ambient temperature was 41.24 °C, and the wall temperature was 46.99 °C. The wall temperature showed a decreasing trend in the following order: top, west, east, environment, bottom, and middle. The surface temperature was higher than the internal temperature, with a larger overall gradient, and the maximum daily temperature difference was 18.6 °C. The temperature diffused from the arris of the west side to the top and lower parts of the wall. The west side temperature showed a decreasing trend in the following order: I7, H7, D8, G7, F7, E7, B8, and A8. The highest temperature was 46.99 °C, and the overall temperature rise showed a lag, with the same delay sequence. The temperatures of different test points on the east side from high to low were I1, H1, E1, C1, F1, G1, D8, B1, and A1. The highest temperature was 44.8 °C. The top temperature was the highest, the middle was in the second, and the bottom temperature was the lowest. In addition, the timing of the highest temperature at the top of the wall had a delay of about 1 h. The timing of the temperature rise in the other wall sensors was consistent with the ambient temperature, and the time at which the highest temperature was reached at the foundation

was 2 h earlier than the ambient temperature. The order of the temperatures of the test points of the wall, from high to low, was I7, I6, I1, I2, I5, I3, and I4. The overall maximum temperature distribution was symmetrical in shape, and the temperature of the west side was higher than that of the east side and the middle. Furthermore, the temperature on the top layer of the east side was consistent with the maximum temperature of the environment. The time at which the highest temperature was reached on the west side of the wall was delayed by 2 h. The order of the temperatures of the test points in the middle of the test wall, from high to low, were E1, E2, E7, E3, E6, E4, and E5. The highest temperature on the east of the wall was higher than the west and middle of the wall, and the overall temperature was lower than the ambient temperature. The time at which the other test points reached their highest temperatures was delayed by 2–4 h in sequence. The order of temperatures at the bottoms of the test points, from high to low, was A8, A1, A7, A2, A6, A3, A5, and A4. The highest temperature on the west side of the bottom was higher than that on the east side, and the overall temperature was lower than the ambient temperature. The time it took to reach the highest temperature on the east side of the wall was 1–2 h lower than that under the ambient temperature, while the time taken to reach the highest temperature on the west side was delayed by 1–2 h.

As can be seen from Figure 22, the lowest ambient temperature occurred on 2018/1/19 (−20.5 °C); the lowest wall temperature was 15.4 °C. Compared with the ambient temperature, the order of the temperatures on each part of the wall, from high to low, was top, west side, east side, bottom, and middle. The surface temperature was lower than that of the interior and had a large overall gradient. The maximum daily temperature difference reached 13.7 °C, and the wall temperature diffused from the edges and corners on the west side of the wall to the top, middle, and lower parts. From low to high, the order of the temperatures of the west test points, from low to high, was I7, H7, G7, F7, E7, D8, C8, B8, and A8. The corresponding order for west test points was I7, H7, G7, F7, E7, D8, C8, B8, and A8. The overall trend gradually increased from top to bottom, which was higher than the lowest ambient temperature. The overall temperature decreased, and the time took to reach the lowest value lagged behind. The delayed time increased in reverse order. The daily temperature difference on the east side was smaller than that on the west side, and the maximum daily temperature difference was 11.3 °C. The order of the temperatures of the top test points, from low to high, was I7, I1, I2, I6, I5, I3, and I4. The overall maximum temperature was symmetrical in shape, and the lowest temperature in the west was lower than that in the east and lower than that in the middle; the top temperature was higher than the lowest temperature of the environment. The overall time taken to reach the lowest temperature was delayed by 1 h. There was no obvious change in the overall temperature of the middle part of the wall, and the overall temperature was higher than the highest ambient temperature. The maximum daily temperature difference was not more than 2 °C, and the order of the overall temperatures of the middle part, from low to high, was E7, E1, E2, E3, E6, E5, and E4. The overall temperature at the bottom of the wall showed no obvious change and was higher than the highest ambient temperature. The maximum daily temperature difference was not more than 3 °C. The order of the bottom temperature of the wall, from low to high, was A8, A1, A7, A2, A3, A6, A5, and A4.

As can be seen from Figures 23 and 24, on the day of the highest ambient temperature of the year, the maximum solar radiation reached 880 W/m$^2$ at 13:00. The sunshine lasted for more than 14 h on this day. In the morning, the sun shone on the east side of the wall, and the temperatures in the middle and lower parts rose to their highest at this moment. The temperature at the top still rose to its highest point at 16:00, in a certain range that was affected by solar radiation. The time at which the highest temperature was reached at the top of the west side was relatively delayed, reaching its peak at 18:00. The time at which the highest temperature was reached in the middle and upper parts was delayed by nearly 1 h, compared to the top. The bottom of the wall was the least exposed area to solar radiation, and compared to its lowest temperature, the top temperature on the west side rose by nearly 14.5 °C, and the top temperature on the east side rose by 12.6 °C. On the day

of the lowest ambient temperature in a year, the solar radiation reached 500 W/m$^2$ at 14:00; the sunshine lasted for more than 10 h on this day. In the morning, the sun shone on the east side. At this time, the temperature rose to its highest in the middle and lower parts of the side wall. The temperature of the top, within a certain range, still increased under the influence of solar radiation, reaching its highest point slightly earlier than 16:00. The time at which the highest temperature was reached on the west side was relatively delayed (17:00). If the bottom of the wall is considered to be the least exposed to solar radiation and based on its lowest temperature, the top temperature on the west side rose by nearly 13.7 °C, and the top temperature on the west side rose by nearly 11.3 °C.

## 5. Discussion

Changes in atmospheric temperature and solar radiation directly affected the temperature field of the wall. As the buried temperature probes were nearly 5 cm away from the outer surface of the wall, the measured wall temperature was lower than the outer surface temperature of the wall. The wall temperature fluctuated with the fluctuations of atmospheric temperature and generally presented a certain lag. The temperature at the top and surface of the wall changed sharply and gradually decreased towards the center. The orders of the overall temperature in spring, summer, autumn, and winter, from high to low, were top, west, east, bottom, and middle. This order was entirely due to the influence of solar radiation.

It can be seen that the annual temperature difference of the wall was not lower than 36 °C, the maximum value was over 62.29 °C, the daily temperature difference was almost unchanged inside the wall, and the maximum daily temperature difference is 24.3 °C in the shallow surface layer of the wall, and the overall temperature of the wall gradually changed by 0.6 °C. Thus, drastic changes in the daily temperature difference of the shallow surface layer would be the main cause of the weathering of the wall surface. Particularly in spring and autumn, the wall surface temperature difference was large, and the wall surface temperature reached its highest value in the summer and the lowest in the winter. The temperature change of the wall was the main driving force for wall surface weathering. The daily temperature difference fluctuates around 0 °C in spring, which was the main external cause of wall deformation affected by the temperature field. In addition, the temperature difference gradient between the inner and outer layers of the wall was large, the temperature difference between different depth layers formed the difference between thermal expansion and cold contraction, and the difference in tensile and compressive stress between the inner layer interface was the external force of the surface exfoliation and pulverization. As can be seen from Figure 27, the temperature fluctuated around 0 °C for 101 days throughout the year. This mainly occurred in the winter (2017/11/10–2018/02/25), when the wall temperature circle was below 0 °C for 15 days. The wall's surface was equivalent to more than 100 freeze-thaw cycles, the inside of the wall was unthawed, and the outside had melted. The volume expansion and internal freeze shrinkage of the shallow layer was the main reason for the weathering and porosity of the shallow layer.

In addition, the sharp change in the daily temperatures at the wall surface, which is caused by heat conduction, heat convection, and heat radiation, is the main reason for wall surface weathering. As can be seen from Figure 28, the daily temperature difference of the atmospheric environment was greater than 20 °C for only 12 days in a year. The west side of the top of the wall was only at this level for 19 days, and the east side of the top was only for two days. The daily temperature difference was more than 10 °C for 286 days, the daily temperature difference on the west side of the top of the wall was more than 10 °C for 274 days, for the east side of the top of the wall this period was 234 days, for the west side of the middle it was only 55 days, for the east side it was 26 days, and there were no days warmer than 10 °C in the walls' foundation. The daily temperature difference was greater than 5 °C for 355 days, for the west side of the top of the wall this period was 349 days, for the east side of the top of the wall it was 346 days, for the west side of the middle it was 290 days, for the east side of the middle it was 257 days, for the west side at the bottom of

the wall it was 118 days, for the east side of the bottom it was 13 days, and for the inside of the wall it was only one day. It can be seen that the days when the temperature difference was greater than 10 °C had the most prominent influence on the wall's surface. There were five times more days when the daily temperature difference was greater than 10 °C at the top of the wall than in the middle, and there were no days when the temperature difference was greater than 10 °C at the bottom. The overall wall temperature changed from 0 to 5 °C, and the overall daily temperature difference was greater than 2 °C. Temperature changes of less than 2 °C in the wall were caused by daily temperature differences. Therefore, a temperature change of less than 2 °C in the wall is the stable range of heat conduction.

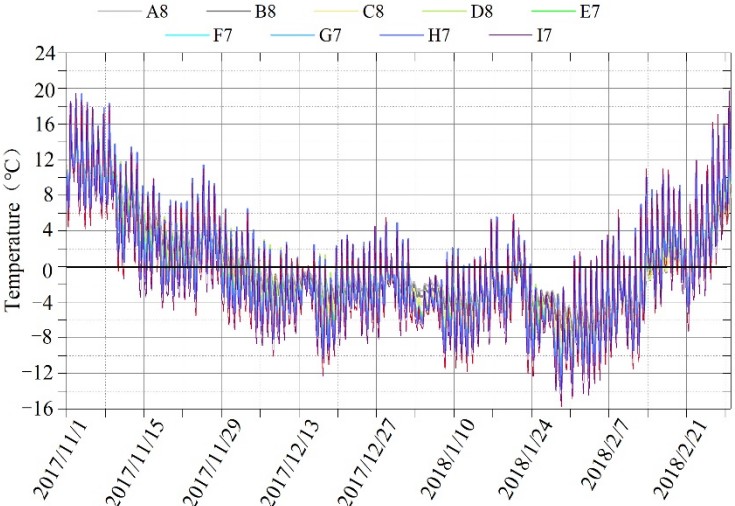

**Figure 27.** Fluctuations at 0 °C.

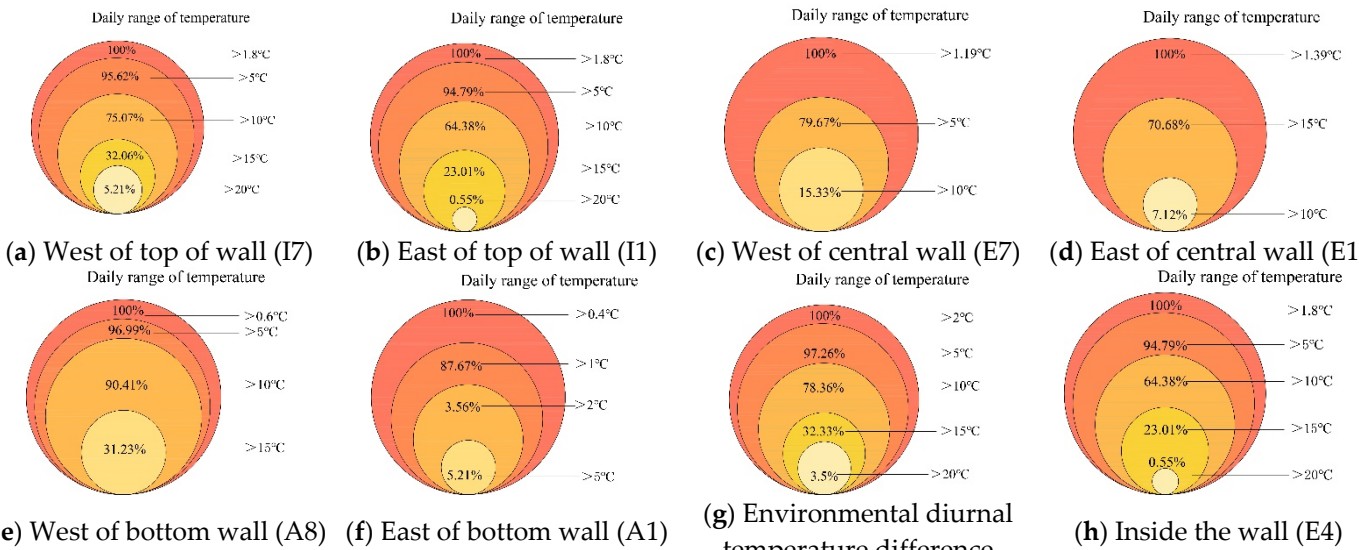

**Figure 28.** Statistical analysis of the daily temperature difference between the atmosphere and the wall.

The influence of ambient temperature changes on the temperature field of the test wall in spring, summer, autumn, and winter is shown in Figure 29. The temperature change on the west part of the wall was greater than that on the east in the sensitive sand transition areas. However, the east part showed greater change than the west part in the stable area. The surface temperature field changed most acutely in summer, followed by spring and autumn; it was relatively stable in winter. The thickness of the shallow surface layer in the

temperature-sensitive area of the test wall was less than 32 cm, and the daily temperature change was more than 10 °C. The temperature change was severe (>30 °C) from 0 to an 18 cm depth of the wall surface. There was an obvious change in depths between 18 and 32 cm (not exceeding 32 cm).

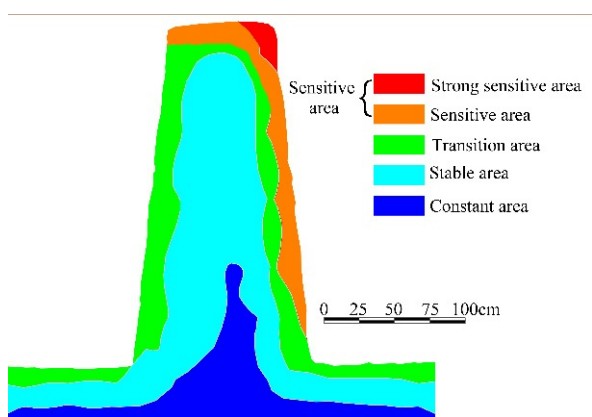

**Figure 29.** Temperature field of test wall across one year.

## 6. Conclusions

Heat conduction, convection, and radiation were identified as the main factors that caused temperature changes in rammed earth walls. The wall surface temperature changed sharply, while the interior remained relatively stable and showed a decreasing gradient, which was the external force for the weathering at the surface of the wall. The temperature field was responsible for the changes to the solid, liquid, and gaseous phases of water, which was the main mode of energy storage and release. Particularly in spring and winter, the alternating temperatures above and below 0 °C were more obviously responsible for the deterioration of the wall surface. The main conclusions were as follows:

(1) The order of the intensity of temperature changes at different positions on the wall, from high to low, was the top west, top east, top center, middle and upper part of the western wall surface, middle and upper part of the eastern wall surface, middle and lower part of the western wall surface; middle and lower part of the eastern wall surface, inner upper side, inner lower side, and finally internal middle. The daily temperature difference on the top surface of the wall was more than 10 °C, accounting for 75% of the days, and the temperature difference inside the wall did not exceed 2 °C. Temperature difference circulation was the main external cause of weathering on the wall surface.

(2) The maximum temperature difference on the surface of the wall was 62.99 °C. The daily gradual change of temperature was not more than 0.6 °C, and the maximum daily temperature difference was 24.3 °C. The changes were the most prominent in spring and autumn, with the highest temperatures occurring in summer and the lowest in winter. The alternating changes above and below 0 °C constituted another main reason for the repeated expansion and contraction weathering of the wall surface, and the extreme point of the annual temperature change was also an important factor.

(3) The temperature-sensitive area of the rammed earth wall was less than 30-cm deep. At 0–18 cm from the edge on the west side of the wall, the temperature change was sharp, whereas the daily temperature difference circulation exceeded 10 °C at depths deeper than 20 cm (the sensitive area). The transition area was 15 cm inside, and the daily temperature difference circulation at this place was 5–10 °C. Finally, in the stable area (10–30 cm inside), the daily temperature difference was 1–5 °C. The rest of the wall was termed as the constant area, whose temperature change was less than 1 °C.

(4) Solar radiation was the main factor that caused the sharp rises and observed lag in the wall temperature. The daily temperature difference between the environment

and the wall was the result of solar radiation. On the day with the highest ambient temperature during the year, the solar radiation reached 880 W/m$^2$ at 13:00 and the sunshine lasted for more than 14 h. On the day with the lowest ambient temperature during the year, the solar radiation was 500 W/m$^2$ at 14:00 and the sunshine lasted for more than 10 h. The daily temperature difference at the top of the west side of the wall was close, which fully showed that the daily temperature difference of the wall was closely related to the angle of solar radiation, the direction of the wall, and radiation time.

(5) The temperature difference in the daily cycle was the main factor that affected the wall deterioration. Over the duration of a year, the order of daily temperature difference, from high to low, was spring, autumn, summer, and winter. The temperature difference was the largest in spring and the most stable in winter. The temperature difference alternated above and below 0 °C in spring, and at this time the volume change because of thermal expansion and cold contraction of rammed soil was the largest. Therefore, spring constituted the main stage of temperature-induced deterioration of the rammed earth wall.

(6) The study revealed the temperature-sensitive areas of rammed earth walls affected by sunlight and environmental temperature, as well as the changes in different seasons, which accumulate experience for researching the deterioration mechanism of earthen sites under the multi-field coupling effect. It is expected to provide the basis for further developing the surface temperature field, stress field and coupling relationship of earthen sites, and provide the basis of environmental action parameters for studying the weathering mechanism of the superficial layer of earthen sites.

**Author Contributions:** Conceptualization, Q.P. and Q.G.; Data curation, B.Z. and J.Z.; Formal analysis, Q.P. and J.H.; Funding acquisition, Q.G.; Investigation, D.S.; Writing—original draft, Q.P.; Writing—review & editing, Q.G. All authors have read and agreed to the published version of the manuscript.

**Funding:** This work was supported by National key research and development plan "Study on Weathering Mechanism and Prevention and Control Technology of Multi-field Coupling Subsoil Sites" (Grant No. 2020YFC1522202); Talent Training and Introduction Program of CAS "Light of the West", Supported by Natural Science Foundation of Gansu Province, "On Dynamic Response Characteristics of Earthen Sites Reinforced by Ramming and Propping"(Grant No. 21JR7RA757).

**Institutional Review Board Statement:** Not applicable.

**Informed Consent Statement:** Not applicable.

**Data Availability Statement:** Not applicable.

**Conflicts of Interest:** The authors declare that they have no known competing financial interest or personal relationships that could have appeared to influence the work reported in this paper.

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
