# Peer review of "Characteristics of Temperature Field of Rammed Earth Wall in Arid Environment"

_coatings, doi:10.3390/coatings12060735_

Round 1
Reviewer 1 Report
The current manuscript describes the characteristics and changes of temperature field of rammed earth wall in arid environment. Minor technical details of the study can be placed in the Supplement Section. The work is interesting and readable. I suggest the following actions for publication:
- Add graphical abstract.
- Figures and tables in the manuscript should be aligned properly.
- Numbering and naming figures and tables should be checked and provide the clear title ( for example, figure 2 it’s not clearly demonstrated the scheme). Avoid the typo errors.
- Figure 6 , PM repeated months in the data , change it.
- What is the significant outcome of this research?
- Is there any optimized temperature did author find.
- Presentation of manuscript is not good enough , The authors did not bother reading the final manuscript. check the manuscript thoroughly and alter it.
- Avoid the typo errors and unnecessary space in the written manuscript.
- Authors must need to incorporate future prospective of the presented work in the conclusion part of the manuscript.
Author Response
Reviewer 1
The current manuscript describes the characteristics and changes of temperature field of rammed earth wall in arid environment. Minor technical details of the study can be placed in the Supplement Section. The work is interesting and readable. I suggest the following actions for publication:
(1)About the question of abstract:Add graphical abstract.
Answer:Graphical abstract has been added and uploaded as attachments.
(2)Figures and tables in the manuscript should be aligned properly.
Answer:Figures and tables in the manuscript revised according to the article format requirements,The specific numbers of the revised figures and tables are as follows:
Fig. 1, Fig. 2, Fig. 3, Fig. 4, Fig. 5, Fig. 6, Fig. 7 ,Fig. 8, Fig. 9, Fig. 10, Fig. 11, Fig. 12, Fig. 13, Fig. 14, Fig. 15, Fig. 16, Fig. 17, Fig. 18, Fig. 19, Fig. 20, Fig. 21 Fig. 22, Fig. 23, Fig. 24,Fig. 25,Fig. 26,Fig. 27,Fig. 27,Fig. 29,Table. 1
(3)Question:Numbering and naming figures and tables should be checked and provide the clear title ( for example, figure 2 it’s not clearly demonstrated the scheme). Avoid the typo errors.
Answer: Numbering and naming figures and tables have been modified. The details are as follows:
- Table 1 headings are numbered and formatted,specifically modified to:
Table. 1 Basic properties of test soils
- Added description to Figure 2
Fig. 2 Testing wall tamping process
- 1- Fig. 23 unify the numbering in the format of 'Fig. x'ï¼›
(4)About the question of Figure 6:Figure 6 , PM repeated months in the data , change it.
Answer: Numbering and naming figures and tables have been modified. The details are as follows:
Fig. 6 Trend charts showing changes in temperature field isotherms over one year
(5)What is the significant outcome of this research?
Answer:This study preliminarily characterized the variation characteristics of 24-hour wall temperature field and the evolution law of four seasons of rammed earthen site in arid environment, discussed the influence of daily and annual temperature difference on the wall, revealed the sensitive area of wall surface temperature and the main seasonal influence characteristics, and put forward that the variation range and amplitude difference of rammed earth wall surface temperature gradient are the main indicators of surface weathering Provide scientific basis for prevention and control of surface weathering.
(6)Is there any optimized temperature did author find.
Answer:Through the study of the wall temperature field, it is found that the highest temperature at the top of the wall was 46.99 ℃, the lowest temperature was −15.3 ℃, and the temperature difference was 62.29 ℃. The highest temperature at the bottom of the wall was 37.3 ℃, the lowest temperature was −5.9 ℃, and the maximum temperature difference was 43.2 ℃. The highest temperature in the middle was 35.8 ℃, the lowest temperature was −5.6 ℃, and the maximum temperature difference was 41.4 ℃. Along the horizontal direction of the wall, the temperature change was more severe in the west than in the east, and it was relatively stable in the middle. The annual maximum temperature difference on the west side was 62.6 ℃, the maximum temperature difference on the east side was 60 ℃, and the maximum temperature difference in the middle was 36.8 ℃. The maximum daily temperature difference of the wall is 24.3 ℃, and the number of days alternating up and down at 0 ℃ accounts for 27.7% of the whole year, with the extreme value of equal temperature gradient change. The results of this study not only provide a scientific basis for the study of the surface weathering mechanism of earthen sites under the coupling action of multiple fields, but also provide a scientific basis for the selection range of environmental simulation parameters and the key areas of site weathering monitoring.
(7)Is there any optimized temperature did author find.
Answer:The author revised the grammar, translation errors and format in the whole article.
(8)Avoid the typo errors and unnecessary space in the written manuscript.
Answer:The author revised the translation errors and unnecessary spaces in the whole article.
(9)Authors must need to incorporate future prospective of the presented work in the conclusion part of the manuscript. Authors must need to incorporate future prospective of the presented work in the conclusion part of the manuscript.
Answer: The research reveals the temperature sensitive area of rammed earth wall affected by sunshine and environmental temperature, as well as the variation characteristics of different seasons, so as to accumulate experience for the prevention and control of surface weathering of rammed earthen site and the study of deterioration mechanism of rammed earthen site under multi field coupling; It is expected to provide reference for the further development of multi field coupling laboratory earthen site full-scale simulation test, the variation law of surface temperature field and stress field, and provide scientific basis for the selection range of environmental simulation parameters and the key areas of site weathering monitoring.

Reviewer 2 Report
The manuscript by Pei et al. investigates the temperature field of rammed earth wall in arid environment. The authors have monitored temperature gradient field of the rammed earth wall from the near-surface to the interior. Their results show that the near surface region is temperature sensitive, but in the interior the temperature does not change substantially.
I recommend publication of this paper subject to the following revisions:
1-Figure 5, panel b: The descriptions in the figure are not readable. Please update the figure.
2-Figure 6: The caption of the figure does not properly describe the figure. Please clearly address both panels of the figure.
3-The horizontal axes of Figures 7, 10, 13, and 16 are not readable. Please update the mentioned figures.
4-I have a technical concern regarding the temperature fluctuations in Fig. 21. The magnitude/amplitude of fluctuations depends on the material of which the wall is constructed, i.e. on the heat capacities (thermal conductivities) of the wall components, and more importantly on the heat conductance/resistance at the boundaries of different materials, which construct the wall (see for example: Polymers 2019, 11, 1465, J. Chem. Theory Comput. 2022, 18, 1870, J. Chem. Phys. 135, 064703, 2011). Of course the authors are not required to report the heat capacities/thermal conductivities, but I recommend the authors to write a few words regarding this so that the readers can better follow the origin of such fluctuations.
Author Response
(1)About the question of Fig. 5: Figure 5, panel b: The descriptions in the figure are not readable. Please update the figure.
Answer: Figure 5, panel b has been modified, as shown in the following figure:
- 5TM sensor layout inside the wall
Fig.5 5TM soil moisture temperature sensor layout
(2)About the question of Figure 6: The caption of the figure does not properly describe the figure. Please clearly address both panels of the figure.
Answer: The title of Figure 6 has been changed to ‘Trend charts showing changes in temperature field isotherms at 4:00 and 16:00 in one year’.
(3)The horizontal axes of Figures 7, 10, 13, and 16 are not readable. Please update the mentioned figures.
Answer:The horizontal axis dates of Figures 7, 10, 13 and 16 have been modified as follows:
|
Fig. 7 Selection of optimum days for spring illumination |
Fig. 10 Selection of optimum days for summer illumination
|
Fig. 13 Selection of optimum days for autumn illumination |
Fig. 16 Selection of optimum days for winter illumination
(4)I have a technical concern regarding the temperature fluctuations in Fig. 21. The magnitude/amplitude of fluctuations depends on the material of which the wall is constructed, i.e. on the heat capacities (thermal conductivities) of the wall components, and more importantly on the heat conductance/resistance at the boundaries of different materials, which construct the wall (see for example: Polymers 2019, 11, 1465, J. Chem. Theory Comput. 2022, 18, 1870, J. Chem. Phys. 135, 064703, 2011). Of course the authors are not required to report the heat capacities/thermal conductivities, but I recommend the authors to write a few words regarding this so that the readers can better follow the origin of such fluctuations.
Answer: The rammed earth wall is made of the same material, and the heat capacities (thermal conductivities) of the same material is unchanged. The wall surface temperature fluctuation is mainly caused by the change of the external environment, including the influence of environmental temperature change, surface heat transfer, heat radiation, etc. the reason for the temperature fluctuation inside the wall is related to the difference of soil heat transfer. The surface heat energy is gradually transferred to the inside and decreases with the increase of depth until it is constant.

Reviewer 3 Report
This is an important paper and reveals an exhaustive collection and processing of data. The introduction fits the theme of the work. The conclusions are interesting.
Given the large number of results, presented for the four seasons of the year, sometimes the reading becomes exhaustive and repetitive, which, in this case, is understandable. There are figures whose reading makes it difficult to prove the results.
The captions for Figures no. 9, 12, 15 and 23 must be improved.
The size of figure 20 makes it difficult to read. Can it be increased?
The inscriptions placed in figure 20 should be more legible
Small lapses:
You must be consistent in the marking of figures. Ex: Fig. x. or Fig. x
In figure 5 you should put parentheses in a) and b)
Line 691 – the name of the newspaper must be lowercase
Line 716 – correct to “ ….of a game …”
Review and standardize the presentation of the bibliography
Author Response
This is an important paper and reveals an exhaustive collection and processing of data. The introduction fits the theme of the work. The conclusions are interesting.
Given the large number of results, presented for the four seasons of the year, sometimes the reading becomes exhaustive and repetitive, which, in this case, is understandable. There are figures whose reading makes it difficult to prove the results.
(1) The captions for Figures no. 9, 12, 15 and 23 must be improved.
Answer: The titles of Figures 9, 12, 15 and 23 have been modified as followsᆪᄎ
Fig. 9 Temperature field of test wall in spring
Fig. 12 Temperature field of test wall in summer
Fig. 15 Temperature field of test wall in autumn
Fig. 18 Temperature field of test wall in winter
Fig. 23 Temperature field of test wall across one year
(2) The size of figure 20 makes it difficult to read. Can it be increased?
Answer: Figure 20 has been rearranged and broken down into figures 20 to 26. The specific modifications are in the graphical abstract and uploaded in the form of attachment.
(3) The inscriptions placed in figure 20 should be more legible
Answer: Figure 20 has been rearranged and broken down into figures 20 to 26. After rearranging, the inscriptions in the figure will be clear. The specific modifications are in the graphical abstract and uploaded in the form of attachment.
(4) Small lapses:
You must be consistent in the marking of figures. Ex: Fig. x. or Fig. x
In figure 5 you should put parentheses in a) and b)
Line 691 - the name of the newspaper must be lowercase
Line 716 - correct to "...of a game..."
Review and standardize the presentation of the bibliography
Answer:
â‘ The picture number has been changed to 'Fig. x' format.
② The numbering of the panels has been changed to 'a) and b) ' formatᆪᄏ
â‘¢ Line 691- the name of the newspaper has been changed to lowercase, and the specific changes are as follows:
[12]DuYumin,Chen Wenwu,Cui Kai,Zhang Jingke,Chen Zhuo,Zhang Qiyong. Damage Assessment of Earthen Sites of the Ming Great Wall in Qinghai Province: A Comparison between Support Vector Machine (SVM) and BP Neural Network[J]. ACM journal on computing and cultural heritage,2020,13(2)
④ Line 716 -"...of a game..."has been changedᆪᆲand the specific changes are as follows:
[24]S. Ye, An interpretation of a game to the moon of the book of rites in terms of comparative mythology exemplified with the state of things in mid-spring, Journal of Shanxi Normal University(Philosophy and Social Sciences Edition) 35(2006)5-10(in Chinese). doi: 10.3969/j.issn.1672-4283.2006.02.001
⑤ The description of non-standard bibliographies in the text has been modified as follows:
[3] M. R. Hall,D. Allinson. Transient numerical and physical modelling of temperature profile evolution in stabilised rammed earth walls[J]. Applied thermal engineering: Design, processes, equipment, economics,2010,30(5)
[32] B. Sun, Z. Zhou, H. Zhang, Y. Zhang, L. Zheng, Characteristics and prediction model of surface temperature for rammed earthen architecture ruins, Rock and Soil Mechanics 3(2011)867-871(in Chinese). doi: 10.3969/j.issn.1000-7598.2011.03.038

Round 2
Reviewer 1 Report
All figures present in manuscript, use same font size and style for the figure title. Few places bold and different font size. Update it.
Author Response
Modification Description
Title:Mechanical characteristics of ramming with a traditional ramming technique
Modification Description:
(1)About the question of figure title:All figures present in manuscript, use same font size and style for the figure title. Few places bold and different font size. Update it.
Answer:In response to question (1), the author has modified the figure title in manuscript to the same font size and style, the details are as follows:
|
|
|
|
a. Rain wash |
|
|
|
|
|
b. Site surface snow |
c. Weathered surface |
|
Fig. 1 Influence of natural conditions |
|
|
|
|
|
|
a) EM50 collector |
b) 5TM soil temperature and humidity sensor |
|
|
Fig. 3 Testing instrument |
||
|
|
|
|
|
a) Plane distribution map of the test area |
b) Environmental data collection |
|
|
Fig. 4 Meteorological station |
||
|
|
|
|
|
a) 5TM sensor is laid out on site |
b) 5TM sensor layout inside the wall |
|
|
Fig.5 5TM soil moisture temperature sensor layout |
||
|
4:00 |
|
|
|
|
|
a) 1 March |
b) 1 April |
c) 1 May |
d) 1 June |
|
|
|
|
|
|
|
|
e) 1 July |
f) 1 August |
g) 1 September |
h) 1 October |
|
|
|
|
|
|
|
|
i) 1 November |
j) 1 December |
k) 1 January |
l) 1 February |
|
|
16:00 |
|
|
|
|
|
m) 1 March |
n) 1 April |
o) 1 May |
p) 1 June |
|
|
|
|
|
|
|
|
q) 1 July |
r) 1 August |
s) 1 September |
t) 1 October |
|
|
|
|
|
|
|
|
u) 1 November |
v) 1 December |
w) 1 January |
x) 1 February |
|
|
|
Fig. 6 Trend charts showing changes in temperature field isotherms at 4:00 and 16:00 in one year |
|||
|
|
|
Fig. 7 Selection of optimum days for spring illumination |
|
|
|
|
|
|
a) 00:00 |
b) 02:00 |
c) 04:00 |
d) 06:00 |
|
|
|
|
|
|
e) 08:00 |
f) 10:00 |
g) 12:00 |
h) 14:00 |
|
|
|
|
|
|
i) 16:00 |
j) 18:00 |
k) 20:00 |
l) 22:00 |
|
Fig. 8 Temperature variations in the test wall on March 22 |
|||
|
|
|
|
|
|
|
|
a) 00:00 |
b) 02:00 |
c) 04:00 |
d) 06:00 |
|
|
|
|
|
|
e) 08:00 |
f) 10:00 |
g) 12:00 |
h) 14:00 |
|
|
|
|
|
|
i) 16:00 |
j) 18:00 |
k) 20:00 |
l) 22:00 |
|
Fig. 11 Temperature variation of test wall on June 16 |
|||
|
|
|
|
|
|
a) 00:00 |
b) 02:00 |
c) 04:00 |
d) 06:00 |
|
|
|
|
|
|
e) 08:00 |
f) 10:00 |
g) 12:00 |
h) 14:00 |
|
|
|
|
|
|
i) 16:00 |
j) 18:00 |
k) 20:00 |
l) 22:00 |
Fig. 14 Temperature variations in the test wall on September 22
|
|
|
|
|
|
a) 00:00 |
b) 02:00 |
c) 04:00 |
d) 06:00 |
|
|
|
|
|
|
e) 08:00 |
f) 10:00 |
g) 12:00 |
h) 14:00 |
|
|
|
|
|
|
i) 16:00 |
j) 18:00 |
k) 20:00 |
l) 22:00 |
Fig. 17 Temperature variations in the test wall on December 22
|
Fig. 18 Temperature field of test wall in winter |
|
|
|
|
|
a) East side of the wall in the year |
b) West side of the wall in the year |
|
Fig. 19 Temperature change characteristic on the wall in the year(2017.4.1–2018.3.31) |
|
|
|
|
|
|
|||||
|
a) Bottom (A1-A8) |
b) Central (E1-E7) |
c) Top (I1-I7) |
|
|||||
|
|
|
|
||||||
|
d) East(A1-I1) |
e) West(A8-I7) |
|
||||||
|
Fig. 20 Wall temperature-atmospheric temperature relationship(2017.4.1–2018.3.31) |
|
|||||||
|
|
|
|
|
|||||
|
a) Bottom (A1-A8) |
b) Central (E1-E7) |
c) Top (I1-I7) |
|
|||||
|
|
|
|
||||||
|
d) East(A1-I1) |
e) West(A8-I7) |
|
||||||
|
Fig. 21 Variation of wall temperature during the highest atmospheric temperature throughout the year(2017.7.13) |
|
|||||||
|
|
|
|
|
|||||
|
a) Bottom (A1-A8) |
b) Central (E1-E7) |
c) Top (I1-I7) |
|
|||||
|
|
|
|
||||||
|
d) East(A1-I1) |
e) West(A8-I7) |
|
||||||
|
Fig. 22 Variation of wall temperature during the lowest atmospheric temperature throughout the year(2018.1.19) |
|
|||||||
|
|
|
|
|
|||||
|
a) Bottom (A1-A8) |
b) Central (E1-E7) |
c) Top (I1-I7) |
|
|||||
|
|
|
|
||||||
|
d) East(A1-I1) |
e) West(A8-I7) |
|
||||||
|
Fig. 23 Diurnal wall temperature change with maximum atmospheric temperature difference in spring(2017.4.22) |
|
|||||||
|
|
|
|
|
|||||
|
a) Bottom (A1-A8) |
b) Central (E1-E7) |
c) Top (I1-I7) |
|
|||||
|
|
|
|
||||||
|
d) East(A1-I1) |
e) West(A8-I7) |
|
||||||
|
Fig. 24 Diurnal wall temperature change with maximum atmospheric temperature difference in autumn(2017.9.6) |
|
|||||||
|
|
|
|
||||||
|
a) East (A1-I1) |
b) West (A8-I7) |
|
||||||
|
Fig. 25 The relationship of wall temperature, atmospheric temperature and the strongest solar radiation(2017.7.13) |
|
|||||||
|
|
|
|
||||||
|
a) East (A1-I1) |
b) West (A8-I7) |
|
||||||
|
Fig. 26 The relationship of wall temperature, atmospheric temperature and the weakest solar radiation(2018.1.19) |
|
|||||||
|
|
|
|
|
|||||
|
a) West of top of wall(I7) |
b) East of top of wall(I1) |
c) West of central wall(E7) |
d) East of central wall(E1) |
|||||
|
|
|
|
|
|||||
|
e) West of bottom wall(A8) |
f) East of bottom wall(A1) |
g) Environmental diurnal temperature difference |
h) Inside the wall(E4) |
|||||
|
Fig. 28 Statistical analysis of the daily temperature difference between the atmosphere and the wall |
||||||||
|
|
||||||||
|
|
||||||||

Reviewer 2 Report
The revised version of manuscript is improved. I recommend publication.
Author Response
Thank you very much for your valuable comments and suggestions, which are of great help to the overall improvement of the paper.